# CAUSAL INFERENCE Q-NETWORK:
# TOWARD RESILIENT REINFORCEMENT LEARNING

## ABSTRACT

Deep reinforcement learning (DRL) has demonstrated impressive performance in various gaming simulators and real-world applications. In practice, however, a DRL agent may receive faulty observation by abrupt interferences such as black-out, frozen-screen, and adversarial perturbation. How to design a resilient DRL algorithm against these rare but mission-critical and safety-crucial scenarios is an important yet challenging task. In this paper, we consider a resilient DRL framework with observational interferences. Under this framework, we discuss the importance of the causal relation and propose a causal inference based DRL algorithm called causal inference Q-network (CIQ). We evaluate the performance of CIQ in several benchmark DRL environments with different types of interferences. Our experimental results show that the proposed CIQ method could achieve higher performance and more resilience against observational interferences.

## 1 INTRODUCTION

Deep reinforcement learning (DRL) methods have shown enhanced performance, gained widespread applications (Mnih et al., 2015; 2016; Ecoffet et al., 2019; Silver et al., 2017; Mao et al., 2017), and improved robot learning (Gu et al., 2017) in navigation systems (Tai et al., 2017; Nagabandi et al., 2018). However, most successful demonstrations of these DRL methods are usually trained and deployed under well-controlled situations. In contrast, real-world use cases often encounter inevitable observational uncertainty (Grigorescu et al., 2020; Hafner et al., 2018; Moreno et al., 2018) from an external attacker (Huang et al., 2017) or noisy sensor (Fortunato et al., 2018; Lee et al., 2018). For examples, playing online video games may experience sudden black-outs or frame-skippings due to network instabilities, and driving on the road may encounter temporary blindness when facing the sun. Such an abrupt interference on the observation could cause serious issues for DRL algorithms. Unlike other machine learning tasks that involve only a single mission at a time (e.g., image classification), an RL agent has to deal with a dynamic (Schmidhuber, 1992) and encoded state (Schmidhuber, 1991; Kaelbling et al., 1998) and to anticipate future rewards. Therefore, DRL-based systems are likely to propagate and even enlarge risks (e.g., delay and noisy pulsed-signals on sensor-fusion (Yurtsever et al., 2020; Johansen et al., 2015)) induced from the uncertain interference.

In this paper, we investigate the *resilience* ability of an RL agent to withstand unforeseen, rare, adversarial and potentially catastrophic interferences, and to recover and adapt by improving itself in reaction to these events. We consider a resilient RL framework with observational interferences. At each time, the agent's observation is subjected to a type of sudden interference at a predefined possibility. Whether or not an observation has interfered is referred to as the interference label.

Specifically, to train a resilient agent, we provide the agent with the interference labels during training. For instance, the labels could be derived from some uncertain noise generators recording whether the agent observes an intervened state at the moment as a binary causation label. By applying the labels as an *intervention* into the environment, the RL agent is asked to learn a binary causation label and embed a latent state into its model. However, when the trained agent is deployed in the field (i.e., the testing phase), the agent only receives the interfered observations but is agnostic to interference labels and needs to act resiliently against the interference.

For an RL agent to be resilient against interference, the agent needs to diagnose observations to make the correct inference about the reward information. To achieve this, the RL agent has to reason about what leads to desired rewards despite the irrelevant intermittent interference. To equip an RL

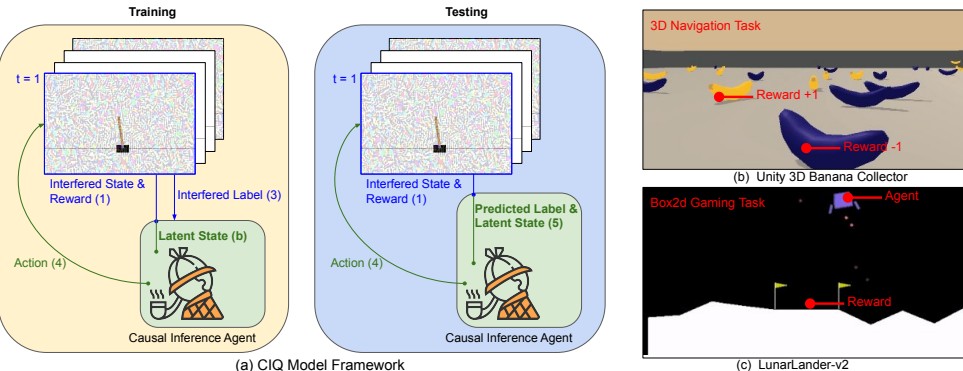

Figure 1: Frameworks of: (a) the proposed causal inference Q-network (CIQ) training and test framework, where the latent state is an unobserved (hidden) confounder; (b) a 3D navigation task, banana collector (Juliani et al., 2018), and (c) a video game, LunarLander (Brockman et al., 2016).

agent with this reasoning capability, we exploit the causal inference framework. Intuitively, a causal inference model for observation interference uses an unobserved confounder (Pearl, 2009; 2019; 1995b; Saunders et al., 2018; Bareinboim et al., 2015) to capture the effect of the interference on the reward collected from the environment.

When such a confounder is available, the RL agent can focus on the confounder for relevant reward information and make the best decision. As illustrated in Figure 1, we propose a causal inference based DRL algorithm termed causal inference Q-network (CIQ). During training, when the interference labels are available, the CIQ agent will implicitly learn a causal inference model by embedding the confounder into a latent state. At the same time, the CIQ agent will also train a Q-network on the latent state for decision making. Then at testing, the CIQ agent will make use of the learned model to estimate the confounding latent state and the interference label. The history of latent states is combined into a causal inference state, which captures the relevant information for the Q-network to collect rewards in the environment despite of the observational interference.

In this paper, we evaluate the performance of our method in four environments: 1) Cartpole-v0 – the continuous control environment (Brockman et al., 2016); 2) the 3D graphical Banana Collector (Juliani et al., 2018)); 3) an Atari environment LunarLander-v2 (Brockman et al., 2016), and 4) pixel Cartpole – visual learning from the pixel inputs of Cartpole. For each of the environments, we consider four types of interference: (a) black-out, (b) Gaussian noise, (c) frozen screen, and (d) adversarial attack.

In the testing phase mimicking the practical scenario that the agent may have interfered observations but is unaware of the true interference labels (i.e., happens or not), the results show that our CIQ method can perform better and more resilience against all the four types of interference. Furthermore, to benchmark the level of resilience of different RL models, we propose a new robustness measure, called CLEVER-Q, to evaluate the robustness of Q-network based RL algorithms. The idea is to compute a lower bound on the observation noise level such that the greedy action from the Q-network will remain the same against any noise below the lower bound. According to this robustness analysis, our CIQ algorithm indeed achieves higher CLEVER-Q scores compared with the baseline methods.

The main contributions of this paper include 1) a framework to evaluate the resilience of DRL methods under abrupt observational interferences; 2) the proposed CIQ architecture and algorithm towards training a resilient DRL agent, and 3) an extreme-value theory based robustness metric (CLEVER-Q) for quantifying the resilience of Q-network based RL algorithms.

## 2 RELATED WORKS

**Causal Inference for Reinforcement Learning:** Causal inference (Greenland et al., 1999; Pearl, 2009; Pearl et al., 2016; Pearl, 2019; Robins et al., 1995) has been used to empower the learning process under noisy observation and have better interpretability on deep learning models (Shalit et al., 2017; Louizos et al., 2017), also with efforts (Jaber et al., 2019; Forney et al., 2017; Bareinboim et al., 2015) on causal online learning and bandit methods. Defining causation and applying causal inference framework to DRL still remains relatively unexplored. Recent works (Lu et al., 2018;

Tennenholtz et al., 2019) study this problem by defining action as one kind of intervention and calculating the treatment effects on the action. In contrast, we introduce causation into DRL by applying extra noisy and uncertain inventions. Different from the aforementioned approaches, we leverage the causal effect of observational interferences on states, and design an end-to-end structure for learning a *causal-observational* representation evaluating treatment effects on rewards.

**Adversarial Perturbation:** An intensifying challenge against deep neural network based systems is adversarial perturbation for making incorrect decisions. Many gradient-based noise-generating methods (Goodfellow et al., 2015; Huang et al., 2017) have been conducted for misclassification and mislead an agent's output action. As an example of using DRL model playing Atari games, an adversarial attacker (Lin et al., 2017; Yang et al., 2020) could jam in a timely and barely detectable noise to maximize the prediction loss of a Q-network and cause massively degraded performance.

**Partially Observable Markov Decision Processes (POMDPs):** Our resilient RL framework can be viewed as a POMDP with interfered observations. Belief-state methods are available for simple POMDP problems (e.g., plan graph and the tiger problem (Kaelbling et al., 1998)), but no provably efficient algorithm is available for general POMDP settings (Papadimitriou & Tsitsiklis, 1987; Gregor et al., 2018). Recently, Igl et al. (2018) have proposed a DRL approach for POMDPs by combining variational autoencoder and policy-based learning, but this kind of methods do not consider the interference labels available during training in our resilient RL framework.

**Safe Reinforcement Learning:** Safe reinforcement learning (SRL) (Garcia & Fernández, 2012) seeks to learn a policy that maximizes the expected return, while satisfying specific safety constraints. Previous approaches to SRL include reward-shaping (Saunders et al., 2018), noisy training (Fortunato et al., 2018), shielding-based SRL (Alshiekh et al., 2018), and policy optimization with confident lower-bound constraints (Thomas et al., 2015). However, finding these policies in the first place could need to reset the model at each time and be computationally challenging. Our proposed resilient RL framework can be viewed as an approach to achieve SRL (Alshiekh et al., 2018), but we focus on gaining resilience against abrupt observation interferences. Another key difference between our framework and other SRL schemes is the novelty in proactively using available interference labels during training, which allows our agent to learn a causal inference model to make safer decisions.

## 3 RESILIENT REINFORCEMENT LEARNING

In this section, we formally introduce our resilient RL framework and provide an extreme-value theory based metric called CLEVER-Q for measuring the robustness of DQN-based methods.

We consider a sequential decision-making problem where an agent interacts with an environment. At each time $t$, the agent gets an observation $x_t$, e.g. a frame in a video environment. As in many RL domains (e.g., Atari games), we view $s_t = (x_{t-M+1}, \ldots, x_t)$ to be the state of the environment where $M$ is a fixed number for the history of observations. Given a stochastic policy $\pi$, the agent chooses an action $a_t \sim \pi(s_t)$ from a discrete action space based on the observed state and receives a reward $r_t$ from the environment. For a policy $\pi$, define the Q-function $Q^\pi(s, a) = \mathbb{E}\left[\sum_{t=0}^\infty \gamma^t r_t | s_0 = s, a_0 = a, \pi\right]$ where $\gamma \in (0, 1)$ is the discount factor. The agent's goal is to find the optimal policy $\pi^*$ that achieves the optimal Q-function given by $Q^*(s, a) = \max_\pi Q^\pi(s, a)$.

### 3.1 RESILIENCE BASE ON AN INTERVENTIONAL PERSPECTIVE

We consider a resilient RL framework where the observations are subject to interference (as illustrated in Fig 1 (a)) as an empirical process in **Rubin's Causal Model (RCM)** (Kennedy, 2016; Holland, 1988; Balke & Pearl, 1997; Robins et al., 2003) for causal inference. Given a type of interference $\mathcal{I}$, the agent's observation becomes:

$$x'_t = F^{\mathcal{I}}(x_t, i_t) = i_t \times \mathcal{I}(x_t) + (1 - i_t) \times x_t \tag{1}$$

where $i_t \in \{0, 1\}$ is the label indicating whether the observation is interfered at time $t$ or not (under the potential outcome estimation (Rubin, 1974)), and $\mathcal{I}(x_t)$ is the interfered observation.

The interfered state is then given by $s'_t = (x'_{t-M+1}, \ldots, x'_t)$. We assume that interference labels $i_t$ follow an i.i.d. Bernoulli process with a fixed interference probability $p^{\mathcal{I}}$ as a noise level. For example, when $p^{\mathcal{I}}$ equals to 10%, each observational state has a 10% chance to be intervened under

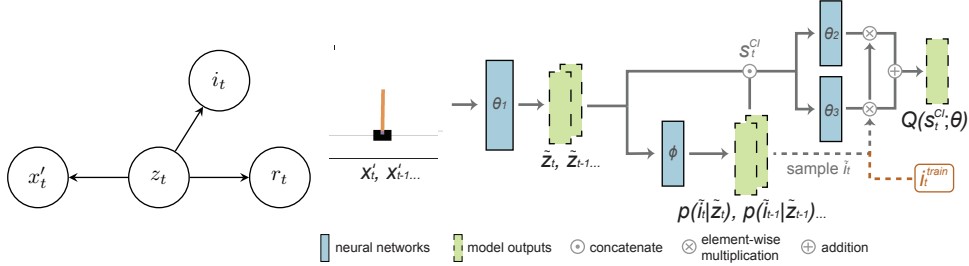

Figure 2: (a) Causal graphical model (CGM). (b) CIQ architecture. The notation $i_t^{train}$ denotes the inference label available during training, whereas $\tilde{i}_t$ is sampled during inference as $i_t$ is unknown.

a perturbation. The agent now needs to choose its actions $a_t \sim \pi(s_t')$ based on the interfered state. The resilient RL objective for the agent is to find a policy $\pi$ to maximize rewards in this environment under observation interference.

In this work, we consider four types of interference as described below.

**Gaussian Noise.** Gaussian noise or white noise is a common interference to sensory data (Osband et al., 2019; Yurtsever et al., 2020). The interfered observation becomes $\mathcal{I}(x_t) = x_t + n_t$ with a zero-mean Gaussian noise $n_t$. The noise variance is set to be the variance of all recorded states.

**Adversarial Observation.** Following the standard adversarial RL attack setting, we use the fast gradient sign method (FGSM) (Szegedy et al., 2014) to generate adversarial patterns against the DQN prediction loss (Huang et al., 2017). The adversarial observation is given by $\mathcal{I}(x_t) = x_t + \epsilon \operatorname{sign}\left(\nabla_{x_t} Q(x_t, y; \theta)\right)$ where $y$ is the optimal output action by weighting over all possible actions.

**Observation Black-Out.** Off-the-shelf hardware can affect the entire sensor networks as a sensing background (Yurtsever et al., 2020) over-shoot with $\mathcal{I}(x_t) = 0$ (Yan et al., 2016). This perturbation is realistic owing to overheat hardware and losing the observational information of sensors.

**Frozen Frame.** Lagging and frozen frame(s) (Kalashnikov et al., 2018) often come from limited data communication bottleneck bandwidth. A frozen frame is given by $\mathcal{I}(x_t) = x_{t-1}$. If the perturbation is constantly present, the frame will remain the first frozen frame since the perturbation happened.

### 3.2 CLEVER-Q: A ROBUSTNESS EVALUATION METRIC FOR Q-NETWORKS

Here we provide a comprehensive score (CLEVER-Q) for evaluating the robustness of a Q-network model by extending the CLEVER robustness score (Weng et al., 2018) designed for classification tasks to Q-network based DRL tasks. Consider an $\ell_p$-norm bounded ($p \geq 1$) perturbation $\delta$ to the state $s_t$. We first derive a lower bound $\beta_L$ on the minimal perturbation to $s_t$ for altering the action with the top Q-value, i.e., the greedy action. For a given $s_t$ and a Q-network, this lower bound $\beta_L$ provides a robustness guarantee that the greedy action at $s_t$ will be the same as that of *any* perturbed state $s_t + \delta$, as long as the perturbation level $\|\delta\|_p \leq \beta_L$. Therefore, the larger the value $\beta_L$ is, the more resilience of the Q-network against perturbations can be guaranteed. Our CLEVER-Q score uses the extreme value theory to evaluate the lower bound $\beta_L$ as a robustness metric for benchmarking different Q-network models. The proof of Theorem 1. is available in appendix B.1.

**Theorem 1.** *Consider a Q-network $Q(s, a)$ and a state $s_t$. Let $\mathcal{A}^* = \arg\max_a Q(s_t, a)$ be the set of greedy (best) actions having the highest Q-value at $s_t$ according to the Q-network. Define $g_a(s_t) = Q(s_t, \mathcal{A}^*) - Q(s_t, a)$ for every action $a$, where $Q(s_t, \mathcal{A}^*)$ denotes the best Q-value at $s_t$. Assume $g_a(s_t)$ is locally Lipschitz continuous[1] with its local Lipschitz constant denoted by $L_q^a$, where $1/p + 1/q = 1$ and $p \geq 1$. For any $p \geq 1$, define the lower bound*

$$\beta_L = min_{a \notin \mathcal{A}^*} g_a(s_t)/L_q^a. \tag{2}$$

*Then for any $\delta$ such that $\|\delta\|_p \leq \beta_L$, we have $\arg\max_a Q(s_t, a) = \arg\max_a Q(s_t + \delta, a)$.*

---

[1]Here locally Lipschitz continuous means $g_a(s_t)$ is Lipschitz continuous within the $\ell_p$ ball centered at $s_t$ with radius $R_p$. We follow the same definition as in (Weng et al., 2018).

### 3.3 Causal Graphical Model and the Benefits on Learning Performance

The causal relation of observation, reward, and interference can be described by a causal graphical model (CGM) in Figure 2. We use $z_t$ to denote the latent state which can be viewed as a *confounder* in causal inference. Formally, we define $z_t = h(x_t, i_t)$ to be the hidden confounder. Here $h$ is a function which compresses $(x_t, i_t)$ into a confounder such that the CGM holds. It is clear from Eq. (1) and the MDP definition that the CGM holds with $h$ being the identity function, i.e., $z_t = (x_t, i_t)$. We assume that there exists some unknown compression function $h$ such that $z_t$ is low-dimensional. Similar to Louizos et al. (2017), we aim to learn to predict this low-dimensional hidden confounder by a neural network.

Table 1: *Causal hierarchy (Pearl, 2009; 2019; Bareinboim et al., 2020) in our resilient DRL setting.*

| Level | Activity | Symbol | Example |
|---|---|---|---|
| ($\mathbb{I}$) Association | Observing | $P(r_t\|x'_t)$ | DQN |
| ($\mathbb{III}$) Intervention | Intervening | $P(r_t\|do(x'_t), i_t)$ | CIQ (ours) |

According to the CGM, different training settings correspond to different levels of Pearl's *causal hierarchy* (Bareinboim et al., 2020; Shpitser & Pearl, 2008; Pearl, 2009) as shown in Table 1. If only the observations are available, the training process corresponds to Level $\mathbb{I}$ of the causal hierarchy, which associates the outcome $r_t$ to the input observation $x'_t$ directly by $P(r_t|x'_t)$. Regular DQN and other algorithms with only observations in training belong to this association level. On the other hand, when interference type $\mathcal{I}$ and the interference labels $i_t$ are available during training, the learning problem is elevated to Level $\mathbb{III}$ of the causal hierarchy. In particular, the interference model of Eq. (1) can be viewed as the intervention logic with the interference label $i_t$ being the treatment information. With these information, we can describe the causal inference problem by

$$P(r_t|do(x'_t), i_t) = P(r_t|F^{\mathcal{I}}(x_t, i_t) = x'_t, i_t) \tag{3}$$

with the do-operator (Pearl, 2019) in the intervention level of the causal hierarchy. Based on the causal hierarchy Theorem (Pearl, 2009), we could answer causal questions in the higher Level $\mathbb{III}$ given the interference type $\mathcal{I}$ and the interference labels $i_t$ in the learning process.

We provide an example to analytically demonstrate the learning advantage of having the interference labels during training. Consider an environment of i.i.d. Bernoulli states with $P(x_t = 1) = P(x_t = 0) = 0.5$ and two actions 0 and 1. There is no reward taking action $a_t = 0$. When $a_t = 1$, the agent pays one unit to have a chance to win a two unit reward with probability $q_x$ at state $x_t = x \in \{0, 1\}$. Therefore, $P(r_t = 1|x_t = x, a_t = 1) = q_x$ and $P(r_t = -1|x_t = x, a_t = 1) = 1 - q_x$. This simple environment is a contextual bandit problem where the optimal policy is to pick $a_t = 1$ at state $x_t = x$ if $q_x > 0.5$, and $a_t = 0$ if $q_x \leq 0.5$. If the goal is to find an approximately optimal policy, the agent should take action $a_t = 1$ during training to learn the probabilities $q_0$ and $q_1$. Suppose the environment is subjected to observation black-out $x'_t = 0$ with $p^{\mathcal{I}} = 0.2$ when $x_t = 1$, and no interference when $x_t = 0$. Assume $q_0 = (3 - q_1)/5$. Then we have $P(r_t = 1|x'_t = 1, a_t = 1) = q_1$, and $P(r_t = 1|x'_t = 0, a_t = 1) = q_0 P(x_t = 0|x'_t = 0) + q_1 P(x_t = 1|x'_t = 0) = 0.5$. If the agent only has the interfered observation $x'_t$, the samples for $x'_t = 0$ are irrelevant to learning $q_1$ because rewards just randomly occur with probability half given $x'_t = 0$. Therefore, the sample complexity bound is proportional to $1/P(x'_t = 1)$ because only samples with $x'_t = 1$ are relevant. On the other hand, if the agent has access to the labels $i_t$ during training, even when observed $x'_t = 0$, the agent can infer whether $x_t = 1$ by checking $i_t = 1$ or not. Therefore, the causal relation allows the agent to learn $q_1$ by utilizing all samples with $x_t = 1$, and the sample complexity bound is proportional to $1/P(x_t = 1) = 2$ which is a 20% reduction from $1/P(x'_t = 1) = 2.5$ when the labels are not available. Note that $z_t = (x_t, i_t)$ is a latent state for this example, and the latent state and its causal relation is very important to improving learning performance.

From the above discussion, we provide the interference type $\mathcal{I}$ the interference labels $i_t$ to efficiently train a resilient RL agent with the CGM; however, in the actual testing environment, the agent only has access to the interfered observations $x'_t$. The CGM allows the agent to infer the latent state $z_t$ and utilize its causal relation with observation, interference and reward to learn resilient behaviors. We will then show how we parameterize this model with a novel deep neural network.

### 3.4 Causal Inference Q-Network

Based on the causal inference graphical model, we propose a causal inference Q-network, referred as CIQ, that is able to map the interfered observation $x'_t$ into a latent state $z_t$, make proper inferences about the interference condition $i_t$, and adjust our policy based on the estimated interference.

We approximate the latent state by a neural network $\tilde{z}_t = f_1(x'_t; \theta_1)$. From the latent state, we generate the estimated interference label $\tilde{i}_t \sim p(\tilde{i}_t|z_t) = f_I(z_t; \phi)$. We denote $s_t^{CI} = (\tilde{z}_{t-M+1}, \tilde{i}_{t-M+1}, \dots, \tilde{z}_t, \tilde{i}_t)$ to be the causal inference state. As discussed in the previous subsection, the causal inference state acts as a confounder between the interference and the reward. Therefore, instead of using the interfered state $s'_t$, the causal inference state $s_t^{CI}$ contains more relevant information for the agent to maximize rewards. Using the causal inference state helps focus on meaningful and informative details even under interference.

With the causal inference state $s_t^{CI}$, the output of the Q-network $Q(s_t^{CI}; \theta)$ is set to be switched between two neural networks $f_2(s_t^{CI}; \theta_2)$ and $f_3(s_t^{CI}; \theta_3)$ by the interference label. Such a switching mechanism prevents our network from over-generalizing the causal inference state. During training, switching between the two neural networks is determined by the training interference label $i_t^{\text{train}}$. We assume that the true interference label is available in the training phase so $i_t^{\text{train}} = i_t$. In the testing, when $i_t$ is not available, we use the predicted interference label $\tilde{i}_t$ as the switch to decide which of the two neural networks to use. The design intuition of the inference mechanism is based on the potential outcome estimation theory (Rubin, 1974; Imbens & Rubin, 2010; Pearl, 2009) in RCM and modeling of the interference scenario as described in Eq. (1). Intuitively, the switching mechanism (counterfactual inference) from RCM could be considered as a method to disentangle a single deep network into two non-parameter-sharing networks to improve model generalization under uncertainty. It has shown many advantages for representation learning in regression tasks (Shalit et al., 2017; Louizos et al., 2017). We provide more implementation details in appendix C.1.

All the neural networks $f_1, f_2, f_3, f_I$ have two fully connected layers[2] with each layer followed by the ReLU activation except for the last layer in $f_2, f_3$ and $f_I$. The overall CIQ model is shown in Figure 2 and $\theta = (\theta_1, \theta_2, \theta_3, \phi)$ denotes all its parameters. Note that, as common practice for discrete action spaces, the Q-network output $Q(s_t^{CI}; \theta)$ is an $\mathcal{A}$-dimensional vector where $\mathcal{A}$ is the size of the action space, and each dimension represents the value for taking the corresponding action.

Finally, we train the CIQ model $Q(s'_t; \theta)$ end-to-end by the DQN algorithm with an additional loss for predicting the interference label. The overall CIQ objective function is defined as:

$$L^{\text{CIQ}}(\theta_1, \theta_2, \theta_3, \phi) = i_t^{\text{train}} \cdot L^{\text{DQN}}(\theta_1, \theta_2, \phi) + (1 - i_t^{\text{train}}) \cdot L^{\text{DQN}}(\theta_1, \theta_3, \phi)$$
$$+ \lambda \cdot (i_t^{\text{train}} \log p(\tilde{i}_t|\tilde{z}_t; \theta_1, \phi) + (1 - i_t^{\text{train}}) \log(1 - p(\tilde{i}_t|\tilde{z}_t; \theta_1, \phi))), \quad (4)$$

where $\lambda$ is a scaling constant and is set to 1 for simplicity. The entire CIQ training procedure is described by Algorithm 1. Due to the design of the causal inference state and the switching mechanism, we will show that CIQ can perform resilient behaviors against the observation interferences.

## 4 Experiments

### 4.1 Environments for DQNs

Our testing platforms were based on (a) OpenAI Gym (Brockman et al., 2016), (b) Unity-3D environments (Juliani et al., 2018), (c) a 2D gaming environment (Brockman et al., 2016), and (d) visual learning from pixel inputs of cart pole. Our test environments cover some major application scenarios and feature discrete actions for training DQN agents with the CLEVER-Q analysis.

**Vector Cartpole:** Cartpole (Sutton et al., 1998) is a classical continuous control problem. The defined environment is manipulated by adding a force of $+1$ or $-1$ to a moving cart. A pendulum starts upright, and the goal is to balance and prevent it from falling over. We use Cartpole-v0 from Gym (Brockman et al., 2016) with a targeted reward $= 195.0$ to solve the environment. The observational vector-state consist of four physical parameters of cart position and angle velocities.

---

[2]Though such manner may lead to the myth of over-parameterization, our ablation study proves that we can achieve better results with almost the same amount of parameters.

---

**Algorithm 1** CIQ Training

---

1: Inputs: $Agent, NoisyEnv, Oracle, max\_step, NoisyEnv\_test, target, eval\_steps$
2: Initialize: $t = 0, score = 0, s'_t = NoisyEnv.$reset()
3: **while** $t < max\_step$ **and** $score < target$ **do**
4:    $i_t = oracle(NoisyEnv, t)$
5:    $a_t = Agent.$act$(s'_t, i_t)$
6:    $s'_{t+1}, r_t, done = NoisyEnv.$step$(a_t)$
7:    $Agent.$learn$(s'_t, a_t, r_t, s'_{t+1}, i_t)$
8:    **if** $t \in eval\_steps$ **then**
9:       $score = Agent.$evaluate$(NoisyEnv\_test)$
10:    **if** $done$ **then**
11:       $s'_t = NoisyEnv.$reset()
12:    **else**
13:       $s'_t = s'_{t+1}$
14:    $t = t + 1$
15: Return $Agent$

---

**Banana Collector:** The Banana collector shown in Figure 1 (b) is one of the Unity baseline (Juliani et al., 2018) rendering by 3D engine. Different from the MuJoCo (Todorov et al., 2012) simulators with continuous actions, the Banana collector is controlled by four discrete actions corresponding to moving directions. The state-space has 37 dimensions included velocity and a ray-based perception of objects around the agent. The targeted reward is 12.0 points by accessing correct bananas ($+1$).

**Lunar Lander:** Similar to the Atari gaming environments, Lunar Lander-v2 (Figure 1 (c)) is a discrete action environment from OpenAI Gym (Brockman et al., 2016). The state is an eight-dimensional vector that records the lander's position, velocity, angle, and angular velocities. The episode finishes if the lander crashes or comes to rest, receiving a reward $-100$ or $+100$ with a targeted reward of 200. Firing ejector costs $-0.3$ each frame with $+10$ for each ground contact.

**Pixel Cartpole:** To further evaluate our models, we conduct experiments from the pixel inputs in the cartpole environment as a visual learning task. The size of input state is $400 \times 600$. We use a max-pooling and a convolution layer to extract states as network inputs. The environment includes two discrete actions $\{left, right\}$, which is identical to the Cartpole-v0 of the vector version.

### 4.2 BASELINE METHODS

In the experiments, we compare our CIQ algorithm with two sets of DRL baselines to demonstrate the resilience capability of the proposed method. We ensure all the models have the **same number** of 9.7 millions **parameters** with careful fine-tuning to avoid model capacity issues.

**Pure DQN:** We use DQN as a baseline in our experiments. The DQN agent is trained and tested on interfered state $s'_t$. We also evaluate common DQN improvements in appendix C.1 and find the improvements have no significant effect against interference.

**DQN with an interference classifier (DQN-CF):** In the resilient reinforcement learning framework, the agent is given the true interference label $i_t^{\text{train}}$ at training. Therefore, we would like to provide this additional information to the DQN agent for a **fair comparision.** During training, the interfered state $s'_t$ is concatenated with the true label $i_t^{\text{train}}$ as the input for the DQN agent. Since the true label is not available at testing, we train an additional binary classifier (CF) for the DQN agent. The classifier is trained to predict the interference label, and this predicted label will be concatenated with the interfered state as the input for the DQN agent during testing.

**DQN with safe actions (DQN-SA):** Inspired by shielding-based safe RL (Alshiekh et al., 2018), we consider a DQN baseline with safe actions (SA). The DQN-SA agent will apply the DQN action if there is no interference. However, if the current observation is interfered, it will choose the action used for the last uninterfered observation as the safe action. This action-holding method is also a typical control approach when there are missing observations (Franklin et al., 1998). Similar to DQN-CF, a binary classifier for interference is trained to provide predicted labels at testing.

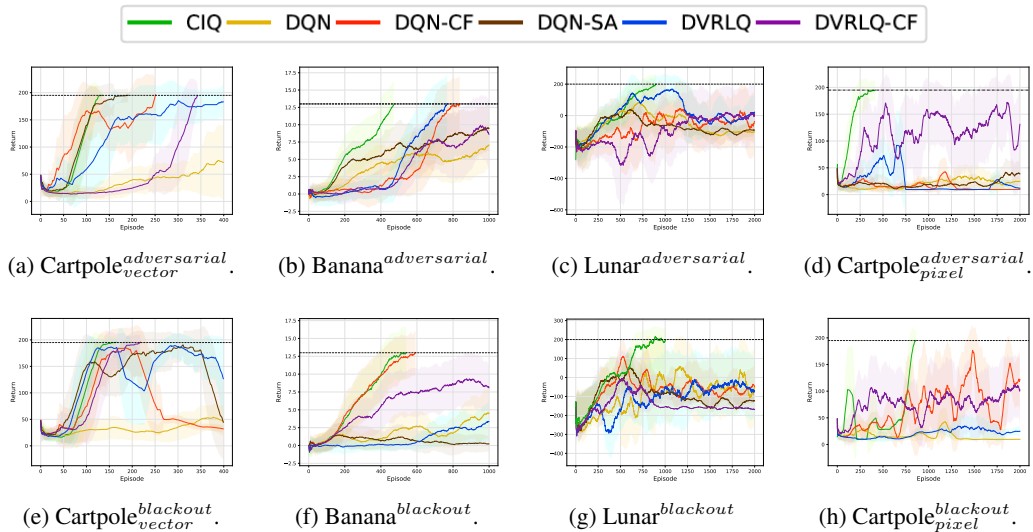

(a) Cartpole$_{vector}^{adversarial}$.  (b) Banana$^{adversarial}$.  (c) Lunar$^{adversarial}$.  (d) Cartpole$_{pixel}^{adversarial}$.

(e) Cartpole$_{vector}^{blackout}$.  (f) Banana$^{blackout}$.  (g) Lunar$^{blackout}$  (h) Cartpole$_{pixel}^{blackout}$.

Figure 3: Performance of DQNs under potential (20%) adversarial and black-out interference.

**DVRLQ and DVRLQ-CF:** Motivated by deep variational RL (DVRL) (Igl et al., 2018), we provide a version of DVRL as a POMDP baseline. We call this baseline DVRLQ because we replace the A2C-loss with the DQN loss. Similar to DQN-CF, we also consider another baseline of DVRLQ with a classifier, referred to as DVRLQ-CF, for a fair comparison using the interference labels.

### 4.3 RESILIENT RL ON AVERAGE RETURNS

We run performance evaluation with six different interference probabilities ($p^I$ in Sec. 3.1), including $\{0\%, 10\%, 20\%, 30\%, 40\%, 50\%\}$. We train each agent 50 times and highlight its standard deviation with lighter colors. Each agent is trained until the target score (shown as the dashed black line) is reached or until 400 episodes. We show the average returns for $p^I = 20\%$ under adversarial perturbation and black-out in Figure 3 and report the rest of the results in appendix C.1.

CIQ (green) clearly outperforms all the baselines under all types of interference, validating the effectiveness of our CIQ in learning to infer and gaining resilience against a wide range of observational interferences. Pure DQN (yellow) cannot handle the interference with 20% noise level. DQN-CF (orange) and DQN-SA (brown) have competitive performance in some environments against certain interferences, but perform poorly in others. DVRLQ (blue) and DVRLQ-CF (purple) cannot achieve the target reward in most experiments and this might suggest the inefficiency of applying a general POMDP approach in a framework with a specific structure of observational interference.

### 4.4 ROBUSTNESS METRICS BASED ON RECORDING STATES

We evaluate the robustness of DQN and CIQ by the proposed CLEVER-Q metric. To make the test state environment consistent among different types and levels of interference, we record the interfered states, $S_N = \mathcal{I}(S_C)$, together with their clean states, $S_C$. We then calculate the average CLEVER-Q for DQN and CIQ based on the clean states $S_C$ using Eq. 2 over 50 times experiments for each agent.

We also consider a retrospective robustness metric, the action correction rate (AC-Rate). Motivated by previous off-policy and error correction studies (Dulac-Arnold et al., 2012; Harutyunyan et al., 2016; Lin et al., 2017), AC-Rate is defined as the action matching rate $R_{Act} = \frac{1}{T} \sum_{t=0}^{T-1} \mathbf{1}_{\{a_t = a_t^*\}}$ between $a_t$ and $a_t^*$ over an episode with length $T$. Here $a_t$ denotes the action taken by the agent with interfered observations $S_N$, and $a_t^*$ is the action of the agent if clean states $S_C$ were observed instead. The roles of CLEVER-Q and AC-Rate are complementary as robustness metrics. CLEVER-Q measures sensitivity in terms of the margin (minimum perturbation) required for a given state to change the original action. AC-rate measures the utility in terms of action consistency. Altogether, they provide a comprehensive resilience assessment.

Table 2 reports the two robustness metrics for DQN and CIQ under two types of interference. CIQ attains higher scores than DQN in both CLEVER-Q and AC-Rate, reflecting better resilience in CIQ evaluations. We provide more robustness measurements in appendix B.2 and E.

Table 2: AC-Rate and CLEVER-Q robustness analysis under Gaussian ($l_2$-norm) and adversarial ($l_\infty$-norm) perturbations in the vector Cartpole environment.

| $\mathcal{I}$=L$_2$ | AC-Rate | | CLEVER-Q | | $\mathcal{I}$=L$_\infty$ | AC-Rate | | CLEVER-Q | |
|---|---|---|---|---|---|---|---|---|---|
| P%, $\mathcal{I}$ | DQN | CIQ | DQN | CIQ | P%, $\mathcal{I}$ | DQN | CIQ | DQN | CIQ |
| 10% | 82.10% | **99.61%** | 0.176 | **0.221** | 10% | 62.23% | **99.52%** | 0.169 | **0.248** |
| 20% | 72.15% | **98.52%** | 0.130 | **0.235** | 20% | 9.68% | **98.52%** | 0.171 | **0.236** |
| 30% | 69.74% | **98.12%** | 0.109 | **0.232** | 30% | 1.22% | **98.10%** | 0.052 | **0.230** |

### 4.5 ADDITIONAL ANALYSIS

We also conduct the following analysis to better understand our CIQ model. Environments with a dynamic noise level are evaluated. Due to the space limit, see their details in appendix C to E.

**Treatment effect analysis:** We provide treatment effect analysis on each kind of interference to statistically verify the CGM with lowest errors on average treatment effect refutation in appendix D.

**Ablation studies:** We conduct ablation studies by comparing several CIQ variants, each without certain CIQ component. The results verify the importance of the proposed CIQ architecture in appendix E.

**Test on different noise levels:** We train CIQ under one noise level and test on another level. The results show that the difference in noise level does not affect much on the performance in appendix C.6.

**Neural saliency map:** We apply the perturbation-based saliency map for DRL (Greydanus et al., 2018) in appendix E.4 to visualize the saliency centers and actions of CIQ and other baseline agents.

**Transferability in robustness:** Based on CIQ, we study how well can the robustness of different interference types transfer between training and testing environments . We evaluate two general settings (i) same interference type but different noise levels (appendix C.6) and (ii) different interference types (appendix E.5).

**Multiple interference types:** We also provide an generalized version of CIQ that deals with multiple interference types at training and testing environments. The generalized CIQ is equipped with a common encoder and individual interference decoders to study multi-module conditional inference, as discussed in appendix E.6.

## 5 CONCLUSION

Our experiments suggest that, although some DRL-based algorithms can achieve high scores under the normal condition, their performance can be severely degraded in the presence of interference. In order to be resilient against interference, we propose CIQ, a novel causal-inference-driven DRL algorithm. Evaluated on a wide range of environments and multiple types of interferences, the CIQ results show consistently superior performance over several RL baseline methods. We also validate the improved resilience of CIQ by CLEVER-Q and AC-Rate metrics and will open source code.

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

## A  APPENDIX

INDEX

Our supplementary sections included:

## B  PROOF OF THE CLEVER-Q THEOREM AND ADDITIONAL ROBUSTNESS MEASUREMENTS

### B.1  PROOF OF THE CLEVER-Q THEOREM

Here we provide a comprehensive score (CLEVER-Q) for evaluating the robustness of a Q-network model by extending the CLEVER robustness score Weng et al. (2018) designed for classification tasks to Q-network based DRL tasks. Consider an $\ell_p$-norm bounded ($p \geq 1$) perturbation $\delta$ to the state $s_t$. We first derive a lower bound $\beta_L$ on the minimal perturbation to $s_t$ for altering the action with the top Q-value, i.e., the greedy action. For a given $s_t$ and a Q-network, this lower bound $\beta_L$ provides a robustness guarantee that the greedy action at $s_t$ will be the same as that of *any* perturbed state $s_t + \delta$, as long as the perturbation level $\|\delta\|_p \leq \beta_L$. Therefore, the larger the value $\beta_L$ is, the more resilience of the Q-network against perturbations can be guaranteed. Our CLEVER-Q score uses the extreme value theory to evaluate the lower bound $\beta_L$ as a robustness metric for benchmarking different Q-network models.

**Theorem 2.** *Consider a Q-network $Q(s, a)$ and a state $s_t$. Let $\mathcal{A}^* = \arg\max_a Q(s_t, a)$ be the set of greedy (best) actions having the highest Q-value at $s_t$ according to the Q-network. Define $g_a(s_t) = Q(s_t, \mathcal{A}^*) - Q(s_t, a)$ for every action a, where $Q(s_t, \mathcal{A}^*)$ denotes the best Q-value at $s_t$. Assume $g_a(s_t)$ is locally Lipschitz continuous[3] with its local Lipschitz constant denoted by $L_q^a$, where $1/p + 1/q = 1$ and $p \geq 1$. Then for any $p \geq 1$, define the lower bound*

$$\beta_L = min_{a \notin \mathcal{A}^*} g_a(s_t)/L_q^a.$$

*Then for any $\delta$ such that $\|\delta\|_p \leq \beta_L$,*

$$\arg\max_a Q(s_t, a) = \arg\max_a Q(s_t + \delta, a)$$

*Proof.* Because $g_a(s_t)$ is locally Lipschitz continuous, by Holder's inequality, we have

$$|g_a(x) - g_a(y)| \leq L_q^a \|x - y\|_p, \tag{5}$$

for any $x, y$ within the $R_p$-ball centered at $s_t$. Now let $x = s_t$ and $y = s_t + \delta$, where $\delta$ is some perturbation. Then

$$g_a(s_t) - L_q^a \|\delta\|_p \leq g_a(s_t + \delta) \leq g_a(s_t) + L_q^a . \|\delta\|_p \tag{6}$$

Note that if $g_a(s_t + \delta) \geq 0$, then $A^*$ still remains as the top Q-value action set at state $s_t + \delta$. Moreover, $g_a(s_t) - L_q^a \|\delta\|_p \geq 0$ implies $g_a(s_t + \delta) \geq 0$. Therefore,

$$\|\delta\|_p \leq g_a(s_t)/L_q^a, \tag{7}$$

provides a robustness guarantee that ensures $Q(s_t + \delta, \mathcal{A}^*) \geq Q(s_t + \delta, a)$ for any $\delta$ satisfying Eq. equation 5. Finally, to provide a robustness guarantee that $Q(s_t + \delta, \mathcal{A}^*) \geq Q(s_t + \delta, a)$ for any action $a \notin \mathcal{A}^*$, it suffices to take the minimum value of the bound (for each $a$) in Eq. equation 5 over all actions other than $a^*$, which gives the lower bound

$$\beta_L = min_{a \notin A^*} g_a(s_t)/L_q^a \tag{8}$$

□

---

[3]Here locally Lipschitz continuous means $g_a(s_t)$ is Lipschitz continuous within the $\ell_p$ ball centered at $s_t$ with radius $R_p$. We follow the same definition as in Weng et al. (2018).

For computing $\beta_L$, while the numerator is easy to obtain, the local Lipschitz constant $L_q^a$ cannot be directly computed. In our implementation, by using the fact that $L_q^a$ is equivalent to the local maximum gradient norm (in $\ell_q$ norm), we use the same sampling technique from extreme value theory as proposed in Weng et al. (2018) for estimating $L_q^a$.

## B.2 Additional Robustness Measurements

Following the discussion in Section 4.4 of the main content, we provide more experimental results related to CLEVER-Q measurement and use action correction rate (AC-Rate) mentioned in the main content as a reference.

Table S1. reports the two robustness metrics for DQN, CIQ, DQN-CF (a DQN joint-training with an interference classifier, denoted as Q-CF), DQN-SA (a DQN joint-training with safe-action replay, denoted as Q-SA), DVRLQ (a deep variational reinforcement learning framework Igl et al. (2018) with a DQN-loss, denoted as V-Q), and DVRLQ-CF (a V-Q joint-training with an interference classifier, denoted as V-CF) under two types of $L_n$-norm Weng et al. (2018) interference. CIQ attains higher scores than DQN in both CLEVER-Q and AC-Rate, reflecting better resilience in CIQ evaluations. The performance of the returns of each agent is shown in Table S2. We observe that variational auto-encoding methods included V-Q and V-CF attaining a lower CLEVER-Q, Act-Rate, and average returns from Table S1 and S2. From previous studies Van Hoof et al. (2016), reasons would be difficulties Van Hasselt et al. (2016) of estimation considering temporal information and the various state is hard to disentangle by a single network from counterfactual learning Shalit et al. (2017); Louizos et al. (2017). We also conduct an experiment on a DQN extension of TD-VAE Gregor et al. (2018), but the performance of this extension becomes even lower in all metrics than the V-CF after carefully fine-tuning. We also find that the DQNs would be a benefit on the performance with a joint-trained interference classifier shown in Table S1 and S2.

Table 3: AC-Rate and CLEVER-Q robustness analysis under Gaussian (Lipschitz $l_2$-norm) and adversarial (Lipschitz $l_\infty$-norm) Huang et al. (2017) perturbations in the vector Cartpole environment.

| $\mathcal{I}=L_2$ | AC-Rate | | CLEVER-Q | | $\mathcal{I}=L_\infty$ | AC-Rate | | CLEVER-Q | |
|---|---|---|---|---|---|---|---|---|---|
| P%, $\mathcal{I}$ | DQN | CIQ | DQN | CIQ | P%, $\mathcal{I}$ | DQN | CIQ | DQN | CIQ |
| 10% | 82.10% | **99.61%** | 0.176 | **0.221** | 10% | 62.23% | **99.52%** | 0.169 | **0.248** |
| 20% | 72.15% | **98.52%** | 0.130 | **0.235** | 20% | 9.68% | **98.52%** | 0.171 | **0.236** |
| 30% | 69.74% | **98.12%** | 0.109 | **0.232** | 30% | 1.22% | **98.10%** | 0.052 | **0.230** |
| P%, $\mathcal{I}$ | Q-CF | Q-SA | Q-CF | Q-SA | P%, $\mathcal{I}$ | Q-CF | Q-SA | Q-CF | Q-SA |
| 10% | 85.10% | 83.58% | 0.185 | 0.182 | 10% | 71.82% | 67.56% | 0.181 | 0.174 |
| 20% | 75.23% | 74.94% | 0.145 | 0.152 | 20% | 55.28% | 51.20% | 0.136 | 0.124 |
| 30% | 72.25% | 71.47% | 0.127 | 0.125 | 30% | 46.45% | 43.27% | 0.109 | 0.102 |
| P%, $\mathcal{I}$ | V-Q | V-CF | V-Q | V-CF | P%, $\mathcal{I}$ | V-Q | V-CF | V-Q | V-CF |
| 10% | 83.65% | 84.32% | 0.184 | 0.188 | 10% | 61.92% | 65.87% | 0.173 | 0.179 |
| 20% | 71.79% | 72.83% | 0.123 | 0.138 | 20% | 51.06% | 53.48% | 0.117 | 0.121 |
| 30% | 69.70% | 70.85% | 0.094 | 0.108 | 30% | 36.99% | 37.89% | 0.087 | 0.092 |

## C Implementation Details and Additional Results

### C.1 Background and Training Setting

To scale to high-dimensional problems, one can use a parameterized deep neural network $Q(s, a; \theta)$ to approximate the Q-function, and the network $Q(s, a; \theta)$ is referred to as the deep Q-network (DQN). The DQN algorithm Mnih et al. (2015) updates parameter $\theta$ according to the loss function:

$$L^{\text{DQN}}(\theta) = \mathbb{E}_{(s_t, a_t, r_t, s_{t+1}) \sim D} \left[ (y_t^{\text{DQN}} - Q(s_t, a_t; \theta))^2 \right]$$

where the transitions $(s_t, a_t, r_t, s_{t+1})$ are uniformly sampled from the replay buffer $D$ of previously observed transitions, and $y_t^{\text{DQN}} = r_t + \gamma \max_a Q(s_{t+1}, a; \theta^-)$ is the DQN target with $\theta^-$ being the target network parameter periodically updated by $\theta$.

Double DQN (DDQN) Van Hasselt et al. (2016) further improves the performance by modifying the target to $y_t^{\text{DDQN}} = r_t + \gamma Q(s_{t+1}, \arg\max_a Q(s_{t+1}, a; \theta); \theta^-)$. Prioritized replay is another DQN improvement which samples transitions $(s_t, a_t, r_t, s_{t+1})$ from the replay buffer according to the probabilities $p_t$ proportional to their temporal difference (TD) error: $p_t \propto |y_t^{\text{DDQN}} - Q(s_t, a_t; \theta)|^\alpha$ where $\alpha$ is a hyperparameter.

We use Pytorch 1.2 to design both DQN and causal inference Q (CIQ) networks in our experiments. Our code can be found in the supplementary material. We use Nvidia GeForce RTX 2080 Ti GPUs with CUDA 10.0 for our experiments. We use the Quantile Huber loss (Dabney et al., 2018) $\mathcal{L}_\kappa$ for DQN models with $\kappa = 1$ in Sup-Eq. 10, which allows less dramatic changes from Huber loss:

$$\mathcal{L}_\kappa(u) = \begin{cases} \frac{1}{2} u^2, & \text{if } |u| \le \kappa \\ \kappa \left( |u| - \frac{1}{2}\kappa \right), & \text{otherwise} \end{cases} \tag{9}$$

The quantile Huber loss (Dabney et al., 2018) is the asymmetric variant of the Huber loss for quantile $\tau \in [0, 1]$ from Sup-Eq. 9:

$$\rho_\tau^\kappa(u) = \left| \tau - \delta_{\{u < 0\}} \right| \mathcal{L}_\kappa(u). \tag{10}$$

After the a maximum update step in the temporal loss $u$ in Sup-Eq. 9, we synchronize $\theta_i^-$ with $\theta_i$ follow the implementation from the OpenAI baseline Dhariwal et al. (2017) in Sup-Eq 11:

$$u_i(\theta_i) = \mathbb{E} \left( \underbrace{y^{\text{DDQN}}}_{\theta_{\text{target}}} - \underbrace{Q(s, a; \theta_i)}_{\theta_{\text{local}}} \right)^2. \tag{11}$$

We use the soft-update Fox et al. (2015) to update the DQN target network as in Sup-Eq 12:

$$\theta_{\text{local}} = \tau \times \theta_{\text{local}} + (1 - \tau) \times \theta_{\text{target}}, \tag{12}$$

where $\theta_{\text{target}}$ and $\theta_{\text{local}}$ represent the two neural networks in DQN and $\tau$ is the soft update parameter depending on the task.

For each environment, in additional to the 5 baselines described in Section 4.2, we also evaluate the performance of common DQN improvements such as deep double Q-networks (DDQN) for DDQN with dueling (DDQN$_d$), DDQN with a prioritized replay (DDQN$_p$), DDQN with a joint-training interference classifier (DDQN-CF), and DDQN with a safe action reply (DDQN-SA). We test each model against four types of interference, Gaussian, Adversarial, Blackout, and Frozen Frame, with $p^I \in [10\%, 20\%, 30\%, 40\%, 50\%]$. We also consider a non-stationary noise-level sampling from a cosine-wave in a range of $[0\%, 30\%]$ for every ten steps. Results in Table S4-S9 show a return averaged from the four types of noise in the $\text{Env}_1$ to $\text{Env}_4$. CIQ shows a better and continuous performance to solve the environments before the noise level attaining 40% and under the cosine-noise. Compared to variational based DQNs methods, joint-trained DDQN-CF show a much obvious advantages when the noise levels are in the range of 40% to 50%.

## C.2 ENV$_1$: CARTPOLE ENVIRONMENT.

We use a four-layer neural network, which included an input layer, two 32-unit wide ReLU hidden layers, and an output layer (2 dimensions). The observation dimension of Cartpole-v1 Brockman et al. (2016) is 4 and the input stacks 4 consecutive observations. The dimension of the input layer is [$4 \times 4$]. We design a replay buffer with a memory of 100,000, with a mini-batch size of 32, the discount factor $\gamma$ is set to 0.99, the $\tau$ for a soft update of target parameter is $5 \times 10^{-3}$, a learning rate for Adam Kingma & Ba (2014) optimization is $5 \times 10^{-4}$, a regularization term for weight decay is $1 \times 10^{-4}$, the coefficient $\alpha$ for importance sampling exponent is 0.6, the coefficient of prioritization exponent is 0.4. We train each model 1,000 times for each case and report the mean of the average final performance (average over all types of interference) in Table S4. $\text{Env}_1$ is often recognized as a simplest environment for DQN training. However, we observe an stability issue of attaining reward over 190.0, when most DQN models attain an over 100.0 score in a 10% noise level. CIQ perform best and competitive results without internal affects from over-parameterization

Table 4: Performance on return in clean and five different noise level in $Env_1$ evaluated by an average of under uncertain perturbation included Gaussian, adversarial, blackout, and frozen frame. All DQN models solve the environment with over 195.0 average returns in a clean state (a.k.a. no noise).

| Model | 0% | 10% | 20% | 30% | 40% | 50% | Cosine | Para. |
|---|---|---|---|---|---|---|---|---|
| DQN | 195.1 | 115.0 | 68.9 | 32.3 | 22.8 | 19.1 | 42.1 | 6.9 M |
| $DDQN_d$ | 195.1 | 130.3 | 85.6 | 57.2 | 29.6 | 23.4 | 68.2 | 9.7 M |
| $DDQN_p$ | 195.1 | 143.6 | 79.3 | 56.1 | 31.2 | 28.6 | 60.1 | 9.7 M |
| CIQ | 195.1 | **195.1** | **195.1** | **195.0** | **168.2** | **113.1** | **195.0** | 9.7 M |
| DQN-CF | 195.1 | 192.4 | 143.6 | 128.7 | 68.2 | 57.8 | 138.3 | 9.7 M |
| DDQN-CF | 195.1 | 190.2 | 153.2 | 138.9 | 71.4 | 54.6 | 141.2 | 10.7 M |
| DVRLQ | 195.1 | 152.6 | 107.8 | 76.12 | 58.12 | 19.2 | 72.9 | 9.7 M |
| DVRLQ-CF | 195.1 | 163.1 | 119.3 | 91.3 | 70.9 | 25.1 | 89.2 | 10.7 M |
| DQN-SA | 195.1 | 143.2 | 132.4 | 112.1 | 82.9 | 23.2 | 107.1 | 10.7 M |
| DDQN-SA | 195.1 | 123.4 | 73.2 | 59.4 | 28.1 | 22.8 | 62.8 | 9.7 M |

Table 5: Ablation study on parameter of different DQN models using in our experiments training under four different noise type of noisy environments ($P = 20\%$), which included Blackout, Adversarial, Gaussian, and Frozen frame for $Env_1$ reported in the main content. The minimal parameters of each model denote as Para. in the Tab. 14.

| Model | Para. | Gaussian | Adversarial | Blackout | Frozen |
|---|---|---|---|---|---|
| DQN | 6.9M | 67.4 | 42.5 | 85.7 | 62.1 |
| CIQ | 9.7M | **195.1** | **195.0** | **195.1** | **195.2** |
| DQN-CF | 9.7M | 149.2 | 129.1 | 161.3 | 167.2 |
| DQN-SA | 9.7M | 128.9 | 144.5 | 109.1 | 165.8 |
| DVRLQ | 9.7M | 107.1 | 87.3 | 127.56 | 142.4 |
| DVRLQ-CF | 10.7M | 112.3 | 97.82 | 131.3 | 152.3 |

## C.3   $ENV_2$: 3D Banana Collector Environment.

We utilize the Unity Machine Learning Agents Toolkit Juliani et al. (2018), which is an open-source[4] and reproducible 3D rendering environment for the task of Banana Collector. A reproducible source code, which is designed to render the collector agent, has been given in the supplementary for both Linux and Windows systems. A reward of $+1$ is provided for collecting a yellow banana, and a reward of $-1$ is provided for collecting a blue banana. We use a six-layer deep network, which includes an input layer, three 64-unit fully-connected ReLU hidden layers, soft-attention layer Rao et al. (2017) (for all DQNs), and an output layer (2 dimensions). We use $[37 \times 4]$ for our input layer, which composes from the observation dimension (37) and the stacked input of 4 consecutive observations. We design a replay buffer with a memory of 100,000, with a mini-batch size of 32, the discount factor $\gamma$ is equal to 0.99, the $\tau$ for a soft update of target parameter is $10^{-3}$, a learning rate for Adam Kingma & Ba (2014) optimization is $5 \times 10^{-4}$, a regularization term for weight decay is $1 \times 10^{-4}$, the coefficient $\alpha$ for importance sampling exponent is 0.6, the coefficient of prioritization exponent is 0.4.

We train each model 1,000 times for each case and report the mean of the average final performance (average over all types of interference) in Table S6. We report the DVRLQ-CF, which attains a higher performance among DVRLQ and DVRLQ-SA, to compare with DQN-CF, DDQN-CF. CIQ still performs an overall best performance compared to the other baselines. Interestingly, learning an interference improves general performance combined with the joint-training frameworks.

## C.4   $ENV_3$: Lunar Lander Environment.

The lunar lander-v2 Brockman et al. (2016) is one of the most challenging environments with discrete actions. The observation dimension of Lunar Lander-v2 Brockman et al. (2016) is 8 and the input stacks 10 consecutive observations. The objective of the game is to navigate the lunar lander

---

[4]Source: https://github.com/Unity-Technologies/ml-agents

Table 6: Performance on return in clean and five different noise level in $Env_2$ evaluated by an average of under uncertain perturbation included Gaussian, adversarial, blackout, and frozen frame. All DQN models solve the environment with over 12.0 average returns in a clean state (a.k.a. no noise).

| Model | 0 % | 10% | 20% | 30% | 40% | 50% | Cosine | Para. |
|---|---|---|---|---|---|---|---|---|
| DQN | 12.0 | 9.5 | 5.1 | 3.4 | 2.0 | 2.1 | 3.2 | 7.6 M |
| DDQN | 12.0 | 10.1 | 5.8 | 6.8 | 4.8 | 2.6 | 6.9 | 9.7 M |
| $DDQN_d$ | 12.0 | 11.8 | 6.7 | 7.4 | 5.4 | 2.8 | 6.6 | 9.7 M |
| $DDQN_p$ | 12.0 | 11.9 | 9.2 | 7.2 | 5.4 | 3.2 | 8.3 | 9.7M |
| CIQ | 12.0 | **12.0** | **12.0** | **12.0** | **11.8** | **7.8** | **12.0** | 9.7 M |
| DQN-CF | 12.0 | **12.0** | 11.5 | 10.1 | 9.0 | 6.9 | 11.8 | 9.7 M |
| DDQN-CF | 12.0 | **12.0** | 11.8 | 10.8 | 10.0 | 7.1 | 11.9 | 10.7 M |
| DVRLQ-CF | 12.0 | **12.0** | 11.2 | 10.5 | 9.2 | 6.8 | 11.6 | 10.7 M |

Table 7: Ablation study on parameter of different DQN models using in our experiments training under four different noise type of noisy environments ($P = 20\%$), which included Blackout, Adversarial, Gaussian, and Frozen frame for $Env_2$ reported in the main content. The minimal parameters of each model denote as Para. in the Tab. 14.

| Model | Para. | Gaussian | Adversarial | Blackout | Frozen |
|---|---|---|---|---|---|
| DQN | 6.9M | 6.1 | 4.5 | 5.2 | 5.7 |
| CIQ | 9.7M | **12.0** | **12.0** | **12.0** | **12.0** |
| DQN-CF | 9.7M | **12.0** | 10.9 | 11.3 | 11.8 |
| DQN-SA | 9.7M | **12.0** | 11.1 | 11.1 | 11.2 |
| DVRLQ | 9.7M | **12.0** | 11.3 | 10.3 | 11.6 |
| DVRLQ-CF | 10.7M | **12.0** | 11.5 | 10.5 | 11.6 |

spaceship to a targeted landing spot without a collision. A collection of six discrete actions controls two real-valued vectors ranging from -1 to +1. The first dimension controls the main engine on and off numerically, and the second dimension throttles from 50% to 100% power. The following two actions represent for firing left, and the last two actions represent for firing the right engine. The dimension of the input layer is [8 × 10]. We design a 7-layers neural network for this task, which includes 1 input layer, 2 layer of 32 unit wide fully-connected ReLU network, 2 layers deep 64-unit wide ReLU networks, 1 soft-attention layer Rao et al. (2017) (for all DQNs), and 1 output layer (4 dimensions). The replay buffer size is 500,000; the minimum batch size is 64, the discount factor is 0.99, the $\tau$ for a soft update of target parameters is $10^{-3}$, the learning rate is $5 \times 10^{-4}$, the minimal step for reset memory buffer is 50. We train each model 1,000 times for each case and report the mean of the average final performance (average over all types of interference) in Table S8. $Env_3$ is a challenging task owing to often receive negative reward during the training. We thus consider a non-stationary noise-level sampling from a cosine-wave in a narrow range of [0%, 20%] for every ten steps. Results suggest CIQ could still solve the environment before the noise-level going over to 30%. For the various noisy test, CIQ attains a best performance over 200.0 the other DQNs algorithms (skipping the table since only CIQ and DQN-CF have solved the environment over 200.0 training with adversarial and blackout interference.)

## C.5   $Env_4$: Pixel Cartpole Environment

To observe pixel inputs of Cartpole-v1 as states, we use a screen-wrapper with an original size of [400, 600, 3]. We first resize the original frame into a single gray-scale channel, [100, 150] from the RGN2GRAY function in the OpenCV. The implementation details are shown in the "pixel_tool.py" and "cartpole_pixel.py", which could be refereed to the submitted supplementary code. Then we stack 4 consecutive gray-scale frames as the input. We design a 7-layer DQN model, which included input layer, the first hidden layer convolves 32 filters of a [8 × 8] kernel with stride 4, the second hidden layer convolves 64 filters of a [4 × 4] kernel with stride 2, the third layer is a fully-connected layer with 128 units, from fourth to fifth layers are fully-connected layer with 64 units, a soft-attention layer Rao et al. (2017) (for all DQNs), and the output layer (2 dimensions). The replay buffer size is

Table 8: Performance on average return in clean and five different noise level in $Env_3$ evaluated by an average of under uncertain perturbation included Gaussian, adversarial, blackout, and frozen frame. All DQN models solve the environment with over 200.0 average returns in a clean state input (a.k.a. no noise).

| Model | 0% | 10% | 20% | 30% | 40% | 50% | Cosine | Para. |
|---|---|---|---|---|---|---|---|---|
| DQN | 200.0 | 54.2 | -102.1 | -134.6 | -200.2 | -298.7 | 12.4 | 12.3M |
| DDQN | 200.0 | 70.1 | 15.8 | -6.8 | -124.8 | -243.2 | 32.5 | 15.5M |
| $DDQN_d$ | 200.0 | 92.2 | 26.7 | -7.4 | -167.4 | -100.1 | 43.3 | 15.5M |
| $DDQN_p$ | 200.0 | 102.1 | 39.2 | -17.2 | -189 | -107.8 | 65.0 | 15.5M |
| CIQ | 200.0 | **200.0** | **200.0** | **107.8** | **50.1** | **17.2** | **200.0** | 15.5M |
| DQN-CF | 200.0 | 98.2 | 81.5 | 40.1 | 9.0 | -59.9 | 85.6 | 15.5M |
| DDQN-CF | 200.0 | 188.2 | 91.8 | 58.2 | 10.0 | -20.1 | 128.9 | 15.5M |
| DVRLQ-CF | 200.0 | 198.2 | 121.1 | 80.2 | 29.2 | 11.2 | 165.4 | 15.5M |

500,000; the minimum batch size is 32, the discount factor is 0.99, the $\tau$ for a soft update of target parameters is $10^{-3}$, the learning rate is $5 \times 10^{-4}$, the minimal step for reset memory buffer is 1000.

We train each model 1,000 times for each case and report the mean of the average final performance (average over all types of interference) in Table S9.

Table 9: Performance on average return in clean and five different noise level in $Env_4$ evaluated by an average of under uncertain perturbation included Gaussian, adversarial, blackout, and frozen frame. Only selected DQN models below solve the environment with over 195.0 average returns in a clean state input (a.k.a. no noise).

| Model | 0% | 10% | 20% | 30% | 40% | 50% | Cosine | Para. |
|---|---|---|---|---|---|---|---|---|
| $DDQN_d$ | 195.0 | 182.7 | 99.7 | 75.5 | 47.4 | 13.2 | 90.4 | 20.5M |
| $DDQN_p$ | 195.0 | 187.9 | 139.2 | 88.3 | 51.6 | 15.9 | 120.3 | 20.5M |
| CIQ | 195.0 | **195.0** | **195.0** | **157.8** | **90.1** | **87.2** | 195.0 | 20.5M |
| DQN-CF | 195.0 | 175.0 | 112.5 | 90.0 | 52.0 | 39.2 | 92.8 | 20.5M |
| DDQN-CF | 195.0 | 187.2 | 162.8 | 117.9 | 59.8 | 49.1 | 102.6 | 20.5M |
| DVRLQ-CF | 195.0 | 193.6 | 121.1 | 80.2 | 63.2 | 41.4 | 91.2 | 20.5M |

Table 10: Ablation study on parameter of different DQN models using in our experiments training under four different noise type of noisy environments ($P = 20\%$), which included Blackout, Adversarial, Gaussian, and Frozen frame for $Env_4$ reported in the main content. The minimal parameters of each model denote as Para. in the Tab. 14.

| Model | Para. | Gaussian | Adversarial | Blackout | Frozen |
|---|---|---|---|---|---|
| DQN | 6.9M | 52.6 | 37.5 | 70.5 | 47.6 |
| CIQ | 9.7M | **195.0** | **195.0** | **195.0** | **195.0** |
| DQN-CF | 9.7M | 119.2 | 99.1 | 131.3 | 137.2 |
| DQN-SA | 9.7M | 98.0 | 114.6 | 79.2 | 135.2 |
| DVRLQ | 9.7M | 107.1 | 87.3 | 127.56 | 142.4 |
| DVRLQ-CF | 10.7M | 121.3 | 97.82 | 131.3 | 152.3 |

## C.6 TRAIN AND TEST ON DIFFERENT NOISE LEVEL

We consider settings with different training and testing noise levels for CIQ evaluation. The (train, test)% case trains with train% noise then tests with test% noise. Their results shown in Table S11 are similar to the cases with the same training and testing noise level. We observe that CIQ have the capability of learning transformable q-value estimation, which attain a succeed score of 195.00 in the noise level $30 \pm 10\%$. Meanwhile, other DQNs methods included DDQN-CF, DVRLQ-CF, DDQN-SA perform a general performance decay in the test on different noise level. This result would be limited to the generalization of power and challenges Bengio (2013); Higgins et al. (2018) in as disentangle unseen state of a single parameterized deep network.

Table 11: Stability test of proposed CIQ (*Train* Noise-Level, *Test* Noise-Level)

| Metrics | (10, 30)% | (30, 10)% | (30, 20)% | (30, 30)% | (30, 40)% | (30, 50)% |
|---------|-----------|-----------|-----------|-----------|-----------|-----------|
| Performance | 182.8 | **195.0** | **195.0** | **195.0** | **195.0** | 185.7 |
| CLEVER-Q | 0.195 | 0.239 | 0.232 | 0.230 | 0.224 | 0.215 |
| AC-Rate | 91.45 % | 98.54% | 98.62% | 99.45% | 98.45% | 92.45% |

## D    CAUSAL EFFECTS

In a causal learning setting, evaluating treatment effects and conducting statistical refuting experiments are essential to support the underlying causal graphical model. Through resilient reinforcement learning framework, we could interpret DQN by estimating the average treatment effect (ATE) of each noisy and adversarial observation. We first define how to calculate a treatment effect in the resilient RL settings and conduct statistical refuting tests including random common cause variable test ($T_c$), replacing treatment with a random (placebo) variable ($T_p$), and removing a random subset of data ($T_s$). The open-source causal inference package Dowhy Sharma et al. (2019) is used for analysis.

### D.1    AVERAGE TREATMENT EFFECT UNDER INTERVENTION

We refine a Q-network with discrete actions for estimating treatment effects based on Theorem 1 in Louizos et al. (2017). In particular, individual treatment effect (ITE) can be defined as the difference between the two potential outcomes of a Q-network; and the average treatment effect (ATE) is the expected value of the potential outcomes over the subjects. In a binary treatment setting, for a Q-value function $Q_t(s_t)$ and the interfered state $\mathcal{I}(s_t)$, the ITE and ATE are calculated by:

$$Q_t^{ITE} = Q_t(s_t)\,(1 - p_t) + Q_t(\mathcal{I}(s_t))p_t \tag{13}$$

$$ATE = \sum_{t=1}^{\mathcal{T}} \frac{\mathbb{E}\left[Q_t^{ITE}(\mathcal{I}(s_t))\right] - \mathbb{E}\left[Q_t^{ITE}(s_t)\right]}{\mathcal{T}} \tag{14}$$

where $p_t$ is the estimated inference label by the agent and $\mathcal{T}$ is the total time steps of each episode. As expected, we find that CIQ indeed attains a better ATE and its significance can be informed by the refuting tests based on $T_c$, $T_p$ and $T_s$.

To evaluate the causal effect, we follow a standard refuting setting Rothman & Greenland (2005); Pearl et al. (2016); Pearl (1995b) with the causal graphical model in Fig. 3 of the main context to run three major tests, as reported in Tab. 13. The code for the statistical test was conducted by Dowhy Sharma et al. (2019), which has been submitted as supplementary material. (We intend to open source as a reproducible result.)

Pearl Pearl (1995a) introduces a "do-operator" to study this problem under intervention. The $do$ symbol removes the treatment $\mathbf{tr}$, which is equal to interference $\mathcal{I}$ in the Eq. (1) of the main content , from the given mechanism and sets it to a specific value by some external intervention. The notation $P(r_t|do(tr))$ denotes the probability of reward $r_t$ with possible interventions on treatment at time $t$. Following Pearl's back-door adjustment formula Pearl (2009) and the causal graphical model in Figure 2 of the main content., it is proved in Louizos et al. (2017) that the causal effect for a given binary treatment $\mathbf{tr}$ (denoted as a binary interference label $i_t$ in Eq. (1) of the main content), a series of proxy variables $\mathbf{X} = (\sum_{t=1}^{\mathcal{T}} x_t) \equiv \mathbf{S}' = (\sum_{t=1}^{\mathcal{T}} s_t')$, as $s_t'$ in Eq. (1) of the main content, a summation of accumulated reward $\mathbf{R} = (\sum_{t=1}^{\mathcal{T}} \mathbf{r_t})$ and a confounding variable $\mathbf{Z}$ can be evaluated by (similarly for $\mathbf{tr} = 0$):

$$p(\mathbf{R}|\mathbf{S}', do(\mathbf{tr} = 1)) = \int_{\mathbf{Z}} p(\mathbf{R}|\mathbf{S}, do(\mathbf{tr} = 1), \mathbf{Z})p(\mathbf{Z}|\mathbf{S}, do(\mathbf{tr} = 1))d\mathbf{Z} \overset{(i)}{=} \int_{\mathbf{Z}} p(\mathbf{R}|\mathbf{S}', \mathbf{tr} = 1, \mathbf{Z})p(\mathbf{Z}|\mathbf{S}')d\mathbf{Z}, \tag{15}$$

where equality (i) is by the rules of do-calculus Pearl (1995a); Pearl et al. (2016) applied to the causal graph applied on Figure 1 of the main content. We extend to Eq. 15 on individual outcome study with DQNs, which is known by the Theorem 1. from Louizos et. al. Louizos et al. (2017) and Chapter 3.2 of Pearl Pearl (2009).

D.2   REFUTATION TEST:

A sampling plan for collecting samples refer to as subgroups (i=1, ..., k). Common cause variation (T-c) is denoted as $\sigma_c$, which is an estimate of common cause variation within the subgroups in terms of the standard deviation of the within subgroup variation:

$$\sigma_c \cong \sum_{i=1}^{k} s_i/k, \qquad (16)$$

where $k$ denotes as the number of sample size. We introduce intervention a error rate $n$, which is a probability to feed error interference (e.g., feed $i_t = 0$ even under interference with a probability of $n$) and results shown in Table 12.

The test (T-p) of replacing treatment with a random (placebo) variable is conducted by modifying the graphical relationship in the proposed probabilistic model in Fig. 3 of the main context. The new assign variable will follow the placebo note but with a value sampling from a random Gaussian distribution. The test of removing a random subset of data (T-r) is to randomly split and sampling the subset value to calculate an average treatment value in the proposed graphical model. We use the official dowhy[5] implementation, which includes: (1) confounders effect on treatment: how the simulated confounder affects the value of treatment; (2) confounders effect on outcome: how the simulated confounder affects the value of outcome; (3) effect strength on treatment: parameter for the strength of the effect of simulated confounder on treatment, and (4) effect strength on outcome: parameter for the strength of the effect of simulated confounder on outcome. Following the refutation experiment in the CEVAE paper, we conduct experiments shown in Tab. S12 and S13 with 10 % to 50 % intervention noise on the binary treatment labels. The results in Tab. S12 show that proposed CIQ maintains a lower rate compared with the benchmark methods included logistic regression and CEVAE (refer to Fig. 4 (b) in Louizos et al. (2017)).

Through Eq. (9) to (10) and the corresponding correct action rate in the main context, we could interpret deep q-network by estimating the average treatment effect (ATE) of each noisy and adversarial observation. ATE Louizos et al. (2017); Shalit et al. (2017) is defined as the expected value of the potential outcomes (e.g., disease) over the subjects (e.g., clinical features.) For example, in navigation environments, we could rank the harmfulness of each noisy observation against q-network from the autonomous driving agent.

Table 12: Absolute error ATE estimate; lower value indicates a much stable causal inference under perturbation on logic direction with $P^I = 10\%$ and $n$=error rate of intervention on the binary label.

| Model | $n$=0.1 | $n$=0.2 | $n$=0.3 | $n$=0.4 | $n$=0.5 |
|-------|---------|---------|---------|---------|---------|
| LR    | 0.062   | 0.084   | 0.128   | 0.151   | 0.164   |
| CEVAE | 0.021   | 0.042   | 0.062   | 0.072   | 0.081   |
| CIQ   | **0.019** | **0.020** | **0.015** | **0.018** | **0.023** |

Table 13: Validation of causal effect by three causal refuting tests. The causal effect estimate is tested by random common cause variable test (T-c), replacing treatment with a random (placebo) variable (T-p – lower is better), and removing a random subset of data (T-r). Adversarial attack outperforms in most tests.

| Noise : $do(\mathcal{I})$ | n = 0.1 | | | | n = 0.2 | | | |
|---------------------------|---------|---------|---------|---------|---------|---------|---------|---------|
| **Method** | ATE | w/ T-c | w/ T-p | w/ T-s | ATE | w/ T-c | w/ T-p | w/ T-s |
| Adversarial | 0.2432 | 0.2431 | 0.0294 | 0.2488 | 0.0868 | 0.0868 | 0.0109 | 0.0865 |
| Black-out | 0.2354 | 0.2212 | 0.0244 | 0.2351 | 0.0873 | 0.0870 | 0.0140 | 0.0781 |
| Gaussian | 0.1792 | 0.1763 | 0.0120 | 0.1751 | 0.0590 | 0.0610 | 0.0130 | 0.0571 |
| Frozen Frame | 0.1614 | 0.1614 | 0.0168 | 0.1435 | 0.0868 | 0.0868 | 0.0140 | 0.0573 |

---

[5]Source:github.com/microsoft/dowhy/causal_refuters

# E  ABLATION STUDIES

## E.1  THE NUMBER OF MODEL PARAMETERS

We also spend efforts on a parameter-study on the results of average returns between different DQN-based models, which included DQN, Double DQN (DDQN), DDQN with dueling, CIQ, DQN with a classifier (DQN-CF), DDQN with a classifier (DDQN-CF), DQN with a variational autoencoder Kingma & Welling (2013) (DQN-VAE), NoisyNet, and using the latent input of causal effect variational autoencoder for Q network (CEVAE-Q) prediction. Overall, CEVAE-Q is with minimal-requested parameters with 14.4 M (in $Env_1$) as the largest model used in our experiments in Tab. 14. CIQ remains roughly similar parameters as 9.7M compared with DDQN, $DDQN_d$, and Noisy Net. Our ablation study in Tab. 14 indicates the advantages of CIQ are not owing to extensive features using in the model according to the size of parameters. CIQ attains benchmark results in our resilient reinforcement learning setting compared to the other DQN models.

Table 14: Ablation study on parameter of different DQN models using in our experiments in $Env_1$, $Env_2$, $Env_3$, and $Env_4$. The minimal parameters of each model denote as Para. in the Table 10.

| Model | Para. | $Env_1$ | $Env_2$ | $Env_3$ | $Env_4$ |
|---|---|---|---|---|---|
| DQN | 6.9M | 20.2 | 3.1 | -113.6 | 10.8 |
| DDQN | 9.7M | 41.1 | 3.5 | -123.4 | 57.9 |
| $DDQN_d$ | 9.7M | 82.9 | 4.7 | -136.3 | 67.2 |
| CIQ | 9.7M | **195.1** | **12.5** | **200.1** | **195.2** |
| DQN-CF | 9.7M | 140.5 | **12.5** | -78.3 | 120.2 |
| DDQN-CF | 12.1M | 161.3 | **12.5** | -10.1 | 128.2 |
| DQN-VAE | 9.7M | 151.1 | 7.6 | -92.9 | 24.1 |
| NoisyNet | 9.7M | 158.6 | 5.5 | 50.1 | 100.1 |
| CEVAE-Q | 12.5M | 39.8 | 11.5 | -156.5 | 45.8 |
| DVRLQ-CF | 10.7M | 107.11 | 9.2 | -34.9 | 42.5 |

Noisy Nets Fortunato et al. (2018) has been introduced as a benchmark whose parameters are perturbed by a parametric noise function. We select Noisy Net in a DQN format as a noisy training baseline with interfered state $s'_t$ concated with a interference label $i_t$ from a classifier.

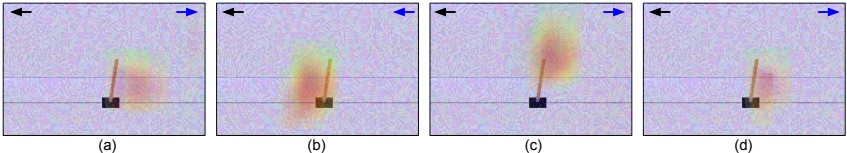

Figure 4: Perturbation-based saliency map on Pixel Cartpole under adversarial perturbation: (a) DQN, (b) CIQ, (c) DQN-CF, and (d) DVRLQ-CF. The black arrows are correct actions and blue arrows are agents' actions. The neural saliency of CIQ makes more correct actions responding to ground actions.

## E.2  LATENT REPRESENTATIONS

We conduct an ablation study by comparing other latent representation methods to the proposed CIQ model.

**DQN with an variational autoencoder (DQN-VAE):** To learn important features from observations, many recent works leverage deep variational inference for accessing latent states for feeding into DQN. We provide a baseline on training a variational autoencoder (VAE) built upon the DQN baseline, denoted as DQN-VAE. The DQN-VAE baseline is targeted to recover a potential noisy state and feed the bottleneck latent features into the Q-network.

**CEVAE-Q Network:** TARNet (Shalit et al., 2017; Louizos et al., 2017) is a major class of neural network architectures for estimating outcomes of a binary treatment on linear data (e.g., clinical reports). Our proposed CIQ uses an end-to-end approach to learn the interventional (causal) features. We provide another baseline on using the latent features from a causal variational autoencoder (Louizos et al., 2017) (CEVAE) as latent features as state inputs followed the loss function in (Louizos et al.,

2017). To get the causal latent model in Q-network, we approximate the posterior distribution by a neural network $z_t \sim p(z_t|\tilde{x}_t) = \phi(\tilde{x}_t; \theta_1)$. Then we train this neural network, CEVAE-Q, by variational inference using the generative model.

We conduct 10,000 times experiments and fine-tuning on DQN-VAE and CEVAE-Q. The results in Table 15 shows that the latent representation learned by CIQ provides better resilience than other representations.

Table 15: Performance on average return in clean and five different noise level in $Env_1$ evaluated by an average of under uncertain perturbation included Gaussian, adversarial, blackout, and frozen frame. All DQN models solve the environment with over 195.0 average returns in a clean state input (a.k.a. no noise).

| Model | 0% | 10% | 20% | 30% | 40% | 50% | Cosine | Para. |
|---|---|---|---|---|---|---|---|---|
| DQN | 195.1 | 115.0 | 68.9 | 32.3 | 22.8 | 19.1 | 42.1 | 6.9 M |
| DDQN | 195.1 | 123.4 | 73.2 | 59.4 | 28.1 | 22.8 | 62.8 | 9.7 M |
| CIQ | 195.1 | **195.1** | **195.1** | **195.0** | **168.2** | **113.1** | **195.0** | 9.7 M |
| DQN-VAE | 195.1 | 173.5 | 141.3 | 124.8 | 86.5 | 33.3 | 101.2 | 9.7 M |
| DQN-CEVAE | 195.1 | 154.4 | 111.9 | 94.8 | 75.5 | 48.3 | 82.1 | 12.5 M |

Table 16: Structure-wise ablation studies of CIQ in $Env_1$ (noise level $P = 20\%$).

| Model | Return | CLEVER-Q | AC-Rate |
|---|---|---|---|
| CIQ | 195.1 | 0.241 | 97.3 |
| B1: CIQ w/o the concatenation | 152.1 | 0.196 | 78.2 |
| B2: CIQ w/o the $\theta_3$ network | 150.1 | 0.182 | 65.6 |
| B3: CIQ w/o providing grounded $i_t$ for training | 135.1 | 0.142 | 53.6 |

### E.3    ARCHITECTURE ABLATION STUDY ON CIQ

To study the importance of specific components in CIQ, we conducted additional ablation studies and constructed two new baseline models shown in Table 16 tested in $Env_1$ (Cartpole). Baseline 1 (B1) - CIQ w/o the concatenation of $\tilde{i}_t$ in $S_I^C$. This comparison shows the importance of using both the predicted confounder $\tilde{z}_t$ and the predicted label $\tilde{i}_t$. B1 uses label prediction to help latent representation but not using the predicted labels in decision-making. The structure is motivated by a task-specific (depth-only information from a maze environment) DQN network from a previous study Mirowski et al. (2016). Baseline 2 (B2) - CIQ w/o the $\theta_3$ network (for testing $\theta_3$'s importance) The structure Humplik et al. (2019) is motivated by a meta-inference reinforcement learning proposed by. Baseline 3 (B3) - CIQ w/o providing grounded $i_t$ for training, for testing the importance of the inference loss and joint loss propagation. The superior performance of CIQ validates the proposed model is indeed crucial from the previous discussion in Section 3 of the main content. The setting used for Table 16 is the same as the setting for the third column (noise level = 20%) in Table 5 and the third column (noise level = 20%) in Table 15, tested in $Env_1$ (Cartpole).

### E.4    PERTURBATION-BASED NEURAL SALIENCY FOR DQN AGENTS

To better understand our CIQ model, we use the benchmark saliency method on DQN agent, perturbation-based saliency map, (Greydanus et al., 2018) to visualize the salient pixels, which are sensitive to the loss function of the trained DQNs. We made a case study of an input frame under an adversarial perturbation, as shown in Fig. 4. We evaluate DQN agents included DQN, CIQ, DQN-CF, DVRLQ-CF and record its weighted center from the neural saliency map, where saliency pixels of CIQ respond to ground true actions more frequent (96.2%) than the other DQN methods.

### E.5    ROBUSTNESS TRANSFERABILITY AMONG DIFFERENT INTERFERENCE TYPES

We conduct additional experiments to study robustness transferability of DQN and CIQ when training and testing under different kinds of interference types in $Env_1$. Note that both architectures would

solve a clean environment successfully (over 195.0). The reported numbers are averaged over 20 independent runs for each condition. As shown in Table 17 and Table 18, CIQ agents consistently attain significant performance improvement when compared with DQN agents, especially between Gaussian and adversarial perturbation. For example, CIQ succeeded to solve the environment 12 times out of 20 independent runs, with an average score of 165.2 in Gaussian (train)-Adversarial (test) adaptation. n particular, for CIQ, 12 times out of 20 independent runs are successfully transfered from Gaussian to Adversarial perturbation. Interestingly, augmenting adversarial perturbation does not always guarantee the best policy transfer when testing in the Blackout and Frozen conditions, which shows a slightly lower performance compared with training on Gaussian interference. The reason could be attributed to the recent findings that adversarial training can undermine model generalization (Raghunathan et al., 2019; Su et al., 2018).

Table 17: DQN adaptation: train and test on different interference (noise level $P = 20\%$) in $\text{Env}_1$.

| Train / Test | Gaussian | Adversarial | Blackout | Frozen |
|---|---|---|---|---|
| Gaussian | 67.4 | 38.4 | 43.7 | 52.1 |
| Adversarial | 53.2 | 42.5 | 35.3 | 44.2 |
| Blackout | 46.2 | 27.4 | 85.7 | 50.3 |
| Frozen | 62.3 | 26.2 | 45.9 | 62.1 |

Table 18: CIQ adaptation: train and test on different interference (noise level $P = 20\%$) in $\text{Env}_1$.

| Train / Test | Gaussian | Adversarial | Blackout | Frozen |
|---|---|---|---|---|
| Gaussian | **195.1** | 165.2 | 158.2 | 167.8 |
| Adversarial | 162.8 | **195.0** | 152.4 | 162.5 |
| Blackout | 131.3 | 121.1 | **195.3** | 145.7 |
| Frozen | 161.6 | 135.8 | 147.1 | **195.2** |

### E.6 CIQ WITH MULTI-INTERFERENCE.

Here we show how the proposed CIQ model can be extended from the architecture shown in Figure 2 to the multi-interference (MI) setting. The design intuition is based on two-step inference by a common encoder, to infer a clean or noisy observation, followed by an individual decoder tied to an interference type, to infer noisy types and activate the corresponding Q-network (named $\theta_4$).

Note that the two-step inference mechanism follows the RCM as two sequential potential outcome estimation models (Rubin, 1974; Imbens & Rubin, 2010), where interfered observation $x'_t$ is determined by two labels $i_{1,t}$ and $i_{2,t}$ according to $x'_t = i_{1,t}(i_{2,t}\mathcal{I}_1(x_t) + (1-i_{2,t})\mathcal{I}_2(x_t)) + (1-i_{1,t})x_t$ extended from Eq.(1), where $i_{1,t}$ indicates the presence of interference and $i_{2,t}$ indicates which interference type (here we show the case of two types). As a proof of concept, we consider two interference types together, Gaussian noise and adversarial perturbation. In this setting every observation (state) can possibly undergo an interference with either Gaussian noise or Adversarial perturbation. From the results shown in Table. 19, we find that the extended version of CIQ, CIQ-MI, is capable of making correct action to solve (over 195.0) the environment when training with mixed interference types (last row). Another finding is that robustness tranferability (153.9/154.2) in CIQ-MI is slightly degraded compared to the results (162.8/165.2) in Table 18 with the same training episodes (500) and runs (20), which could be caused by the increased requirement of model capacity (Ammanabrolu & Riedl, 2019) of CIQ-MI.

Table 19: CIQ-MI: CIQ agent with an extended multi-interference (MI) architecture testing in $\text{Env}_1$ (noise level $P = 20\%$).

| Train / Test | Gaussian | Adversarial | Gaussian + Adversarial |
|---|---|---|---|
| Gaussian | **195.1** | 154.2 | 96.3 |
| Adversarial | 153.9 | **195.0** | 105.1 |
| Gaussian + Adversarial | **195.0** | **195.0** | **195.0** |

