# OpenReview forum: "Causal Inference Q-Network: Toward Resilient Reinforcement Learning"
_ICLR.cc/2021/Conference — Reject_

### Official Review · AnonReviewer1 · 2020-10-26
**Good experimental results but approach lacking in motivation and explanation**

**Rating:** 4
**Confidence:** 4

**Review:**

This paper proposes a method, the Causal Inference Q-Network (CIQ), for training deep RL agents that are robust to abrupt interferences in observations, such as frame blackouts, Gaussian noise or adversarial perturbations. During training time, a binary interference label is provided to the agent at each time step indicating whether an interference has been applied to the observation; the interference label acts as a switch between two neural networks that process the observation to predict the Q-values. The CIQ agent learns to predict the interference label in a supervised fashion, and at test time it uses the predicted label to switch between networks, rather than the true label. The CIQ agent is shown to learn faster and more effectively when compared to a number of baselines on a selection of OpenAI Gym tasks that are modified to include various types of observational interferences.

While this paper presents a method that is shown to perform well empirically in the setting is aimed to tackle, I cannot recommend it for acceptance because (i) almost no motivation or intuition is given for the architectural choices that seem to be key to the performance of the agent (in particular the switching mechanism between Q-networks), and (ii) the characterisation of the agent as performing causal inference, which is the key message of the paper, is confusing in a number of ways. As a result, it’s not clear what we are supposed to learn conceptually from the paper and how it can guide future research.

Positives:
	•	In order to be able to apply deep RL agents in real-world settings, they need to be robust to noisy observations, and so the paper is tackling an important problem.
	•	The chosen baselines seem fairly chosen, such as the safe-action DQN and the DQN with concatenated interference label, and the agents are evaluated across a variety of different interference types and levels. A number of different metrics are used to evaluate the performance of each agent.
	•	The CIQ agent performs the best when compared to the baselines in the interfered observation setting.

Concerns:
	•	There is little to no motivation given for the architectural choice of having the interference label modulate a switching mechanism between Q-networks, which based on the comparison to baselines and the ablation studies in Appendix E.3 seem to be key to the agent’s performance. The only justifying claim I could identify was the following: “Such a switching mechanism prevents our network from over-generalizing the causal inference state.” Could the authors clarify what exactly they mean by over-generalizing the causal inference state and how can this be inferred from the empirical results?
	•	Both the CIQ and DQN-CF agent learn to predict the interference probability and use it for predicting Q-values, with the difference that the CIQ uses this predicted probability to switch between two Q-networks, but DQN-CF performs significantly worse on average - what is the intuition for this?
	•	The causal graph drawn in Figure 2a seems to contradict the experimental setup described in the text:
	◦	(i) The interference label i_t is stated to be sampled from a Bernoulli process but in the causal graph drawn in Figure 2a, there is an arrow from the latent state z_t towards i_t; in what way does the latent state affect the probability of interference?
	◦	(ii) In the graph, there is no arrow (and no indirect path) from the interference label to the observed state x’_t; this implies that intervening on the label, which according to Equation 1 directly affects the value of x’_t, would not actually affect x’_t. This is directly contradictory.
	◦	(iii) In the graph, there is an arrow from i_t to r_t but there is no mention in the paper for how the interference label can directly affect the reward in a given time step.
It’s possible I have badly misunderstood the translation of the setup into the causal graph provided - in which case, could the authors please explain where I have gone wrong?
	•	Section 3.3 describes a toy example that is used to show how being provided interference labels during training can lead to better sample efficiency in learning the reward distributions for each of two states (one of which is occasionally subject to interference that switches the observation to the other state). In this example, however, it is not possible to infer the latent state from the potentially interfered observation at test time, while in the DQN agents the advantage of the CIQ agent is stated to come from the fact that it can do just that. It’s not clear here whether the message is that the CIQ agent performs better due to better sample efficiency or due to its ability to infer the latent state at test time.

Other questions / comments
	•	 As far as I understood, the interference at test time always corresponds to the interference provided during training in the experiments. Were any experiments run to test generalisation to unseen types of interferences at test time? In a real world setting, this could be useful.

---

> ### Author Response · Authors · 2020-11-20
> **Response to AnonReviewer1 (1/2)**
>
> Thank you for recognizing the defined problem in this work is important, the acknowledgment of experiment results and the performance discussion of CIQ. We apologize for the challenges you experienced when reading our initial version. As described in our general response, we have incorporated the review comments and improved our presentation accordingly. Below we provide detailed responses according to the reviewer’s comments.
>
> ****
>
> **R1Q1** - “There is little to no motivation given for the architectural choice of having the interference label modulate a switching mechanism between Q-networks”
>
> **Ans** -  Following your comment, we include a paragraph in Section 3.4 of the revised paper to elaborate on the motivation for the architectural choice (see general response).
>
> ****
>
> **R1Q2** - “  Both the CIQ and DQN-CF agent learn to predict the interference probability and use it for predicting Q-values, with the difference that the CIQ uses this predicted probability to switch between two Q-networks, but DQN-CF performs significantly worse on average - what is the intuition for this?”
>
> **Ans** - Not propagating the interference loss to A-network is one key difference between CIQ and DQN-CF, but CIQ also has other architectural innovations as following explanations. Predicting the interference type is indeed a key feature of CIQ, but the CIQ architecture is different from the regular DQN by incorporating causal inference insight from Rubin's Causal Model (RCM) (Rubin, 1974; Imbens & Rubin, 2010). We also motivate the design mechanism from both recent works from causal representation learning models and theoretical foundations in the updated section 3.3. and 3.4 (see the general response (1) and (2)).  Intuitively, the switching mechanism (counterfactual inference) from RCM could be considered as a method to disentangle a single deep network into two non-parameter-sharing networks to improve model generalization under uncertainty. It has shown many advantages for representation learning in regression tasks (Shalit et al., 2017; Louizos et al., 2017). We show in our ablation study (Appendix E.3) that this mechanism using the predicted label is important for performance.
>
> ****
>
> **R1Q3** - ”The causal graph drawn in Figure 2a seems to contradict the experimental setup described in the text:  (i) The interference label i_t is stated to be sampled from a Bernoulli process but in the causal graph drawn in Figure 2a, there is an arrow from the latent state z_t towards i_t; in what way does the latent state affect the probability of interference?  ◦ (ii) In the graph, there is no arrow (and no indirect path) from the interference label to the observed state x’_t; this implies that intervening on the label, which according to Equation 1 directly affects the value of x’_t, would not actually affect x’_t. This is directly contradictory.  ◦ (iii) In the graph, there is an arrow from i_t to r_t but there is no mention in the paper for how the interference label can directly affect the reward in a given time step.  It’s possible I have badly misunderstood the translation of the setup into the causal graph provided - in which case, could the authors please explain where I have gone wrong? “
>
> **Ans** - To clarify details of our causal model, we add a new paragraph in Section 3.3 (see the general response (1)). The latent state is defined by $z_t = h(x_t, i_t)$, so when $h$ is the identity function, the arrows discussed in points (i) and (ii) are valid because $z_t = (x_t, i_t)$ clearly affects the interference label $i_t$ and the observed state $x’_t$. We assume that there exists $h$ which compresses $z_t$ to a low-dimensional confounder such that the CGM holds, and we aim to learn to predict the confounder via a neural network. For point (iii), we originally draw an arrow from interference to reward because one could consider the causal relation of a more general resilient setting where the reward is also subjected to interference. But since the interference only affects observation in the considered setting as the reviewer correctly pointed out, we remove the arrow from interference to reward in the revised version to avoid confusion. Note that having this arrow or not doesn’t affect the causal inference process described in Section 3.3.

---

> > ### Author Response · Authors · 2020-11-20
> > **Response to AnonReviewer1 (2/2)**
> >
> > **R1Q4** - “Section 3.3 describes a toy example that is used to show how being provided interference labels during training can lead to better sample efficiency in learning the reward distributions for each of two states (one of which is occasionally subject to interference that switches the observation to the other state). In this example, however, it is not possible to infer the latent state from the potentially interfered observation at test time, while in the DQN agents the advantage of the CIQ agent is stated to come from the fact that it can do just that. It’s not clear here whether the message is that the CIQ agent performs better due to better sample efficiency or due to its ability to infer the latent state at test time.”
> >
> > **Ans.** - It’s correct that it’s not possible to precisely know the actual latent state $z_t$ at test time. However, the CIQ agent can infer an estimated $\tilde{z}_t$ for $z_t$ at test time. Using $\tilde{z}_t$ with the training labels allow the agent to have better learning efficiency. To verify the learning efficiency with training labels, we conduct an additional ablation study to compare results using the same CIQ architecture but trained either without or without training labels (baseline model B3 in Table G4.) in appendix E.3. The ablation study shows the benefits and importance of using training labels. Note that CIQ utilizes the benefits of training labels for causal inference as well as the architectural innovation discussed in our response to your previous question (R1Q2).
> >
> > Table. G4. A new baseline in architecture ablation study of CIQ
> >
> > | Model                                             | Return | CLEVER-Q | AC-Rate |
> > |---------------------------------------------------|--------|----------|---------|
> > | CIQ                                               | 195.1  | 0.241    | 97.3    |
> > | B3: CIQ w/o providing grounded $i_t$ for training | 135.1  | 0.142    | 53.6    |
> >
> > ****
> >
> > **R1Q5** - As far as I understood, the interference at test time always corresponds to the interference provided during training in the experiments. Were any experiments run to test generalization to unseen types of interferences at test time? In a real-world setting, this could be useful.
> >
> > **Ans.** - Following the reviewer’s suggestion, we conduct additional experiments to study the robustness transferability of  DQN  and  CIQ  when training and testing under different kinds of interference types in Env1. Please see  Table G2&G3 and the discussion in appendix E.5. Note that both architectures would solve a clean environment successfully (over 195.0).  The reported numbers are averaged over 20 independent runs for each condition.  As shown in Table G2&G3, CIQ agents consistently attain significantly improved performance when compared the averaged returns with  DQN agents, especially between Gaussian (train) and adversarial perturbation (test). In particular, for CIQ, 12 times out of 20 independent runs are successfully transferred from Gaussian to Adversarial perturbation. Interestingly, augmenting adversarial perturbation does not always guarantee the best policy transfer when testing in the Blackout and Frozen conditions, which shows a slightly lower performance compared with training on Gaussian interference. The reason could be attributed to the recent findings that adversarial training can undermine model generalization (A. Raghunathan 2019; Su 2018).
> >
> > Table. G2. DQN adaptation: train and test on different interference (noise) in Env$_1$.
> >
> > | Train\Test | Gaussian | Adversarial| Blackout| Frozen |
> > |:-----------------------|:-------------------:|:----------------------:|:-------------------:|-----------------:|
> > | Gaussian              | 67.4              | 38.4                 | 43.7              | 52.1            |
> > | Adversarial           | 53.2              | 42.5                 | 35.3              | 44.2            |
> > | Blackout              | 46.2              | 27.4                 | 85.7              | 50.3            |
> > | Frozen                | 62.3              | 26.2                 | 45.9              | 62.1            |
> >
> > Table G3. CIQ adaptation: train and test on different interference (noise) in Env$_1$.
> >
> > | Train\Test | Gaussian| Adversarial| Blackout | Frozen|
> > |-----------------------|-------------------|----------------------|-------------------|-----------------|
> > | Gaussian              | 195.1    | 165.2                | 158.2             | 167.8           |
> > | Adversarial           | 162.8             | 195.0       | 152.4             | 162.5           |
> > | Blackout              | 131.3             | 121.1                |195.3    | 145.7           |
> > | Frozen                | 161.6             | 135.8                | 147.1             | 195.2  |

---

> > ### Comment · AnonReviewer1 · 2020-11-20
> > **Still confused about CGM in Figure 2a**
> >
> > Dear authors, thanks very much for your response. I am still confused why there is no arrow or indirect path going from i_t to x'_t in the CGM in Figure 2a given that x'_t is defined to be a *function* of i_t in Equation 1. The CGM would imply that if you were to intervene on i_t it would have no effect on x'_t, directly contradicting Equation 1, which describes the causal mechanism via which i_t influences x'_t. Where have I gone wrong here?

---

> > > ### Author Response · Authors · 2020-11-20
> > > **Clarification about CGM in Figure 2a**
> > >
> > > Thank you for the question.
> > >
> > > - If there is no the confounder $z_t$, there should indeed be an arrow from $i_t$ to $x'_t$ due to their causal relation. In our model, however, we explicitly introduce a confounding variable $z_t$ to capture the causal relation among the variables. When $z_t = (x_t, i_t)$, we can write $x'_t = F^I(z_t)$, and $i_t = g(z_t)$ where $g(z_t)$ outputs the second component of $z_t$. Then, it's clear that both $x'_t$ and $i_t$ are affected by $z_t$, and $x'_t$ and $i_t$ are independent conditioned on $z_t$, i.e., $P(x'_t, i_t | z_t) = P(x'_t | z_t) P(i_t | z_t)$.
> > >
> > > -  When $z_t = h(x_t, i_t)$, our assumption means that $x'_t = F^I(z_t)$ and $i_t = g(z_t)$ hold for some appropriate functions $F^I$ and $g$, so the conditional independence between $x'_t$ and $i_t$ is still true, which leads to the CGM in Figure 2a. We do make use of the conditional independence relation of $x'_t$ and $i_t$ conditioned on $z_t$ in our CIQ model so that we first predict $\tilde z_t$ from $x'_t$, and then use $\tilde z_t$ to predict $\tilde i_t$.
> > >
> > > Hope the above explanation answers your question. Thank you again.

---

> > > > ### Author Response · Authors · 2020-11-24
> > > > **Does the reviewer find our clarification useful?**
> > > >
> > > > Dear Reviewer 1,
> > > >
> > > > Given the fact that the author rebuttal phase will be ending today, we would like to know does our reply address your question on the causal graphical model? We are happy to have more discussion on this matter.

---

> > > > > ### Comment · AnonReviewer1 · 2020-11-24
> > > > > **Yes but still a little confused**
> > > > >
> > > > > Dear authors, thanks for your clarification - I think I understand a bit better now but I am confused how to interpret Equation 3 in context of the causal graph in Figure 2a. It seems to me that the do intervention on $x'_t$ will have no effect on $r_t$ since it is blocked by $z_t$. Or does Equation 3 refer to the causal graph before $z$ is introduced?

---

> > > > > > ### Author Response · Authors · 2020-11-24
> > > > > > **Follow-up clarification**
> > > > > >
> > > > > > We understand the reviewers' confusion, and we are happy to clarify with more details.
> > > > > >
> > > > > > ****
> > > > > >
> > > > > > The reviewer is correct that the $do$ intervention on $x^\prime_t$ does not affect the current reward $r_t$ according to Fig. 2a. However, we note that, once the observation $x^\prime_t$ is intervened, the agent's action will be affected according to the policy. Therefore, through the agent's action, the subsequent confounders ($z_{t+1}, z_{t+2}$, .... etc) will be affected by $x^\prime_t$ and the do-operation at time $t$, which will in turn affect future rewards ($r_{t+1}, r_{t+2}$, .... etc) according to Fig. 2a.
> > > > > >
> > > > > > Note that the training objective is to maximize the Q-function, which is given by $\sum_{k=0}^\infty \gamma^k r_{t + k}$ for the agent at time $t$. As a result, the current intervention at time $t$ would affect the Q-function, which is exactly the value the CIQ agent is predicting with the help of the causal knowledge.
> > > > > >
> > > > > > ****
> > > > > >
> > > > > > We appreciate the reviewer for the in-depth discussion again.

---

### Official Review · AnonReviewer3 · 2020-10-28
**Novel framework but much to do**

**Rating:** 7
**Confidence:** 4

**Review:**

Paper summary:
The paper makes two main contribution: 1). A metric for evaluating RL agent's relience 2). A framework that uses inteference type as label to enhance relience.

Reasons for score:
Overall, I am towards accepting the paper but I believe there are much improvements to make. The idea of adding interference type as label is quite novel, and the authors provide extensive experiment results to show that CIQ achieves better resilience against interferences. I believe that the proposed framework has the potential to enhance a single agent with multiple types of attcks and I suggest the authors to look into this direction.

Comments:
- The proposed network architecture does not seem to differ from a normal DQN other than an additional interference type output. As such, I am not very convinced by the causal inference insight.
- Is it possible to have multiple interference type during training? The resulting agent would then have the potential to be resilient to all types of interferences during evaluation. Furthermore, it would then be possible to combine different types of adversarial attacks into adversarial training. For real life application, I believe that it is possible that different types of interference could happen simultaneously.
- Computation cost of CLEVER-Q is expensive, and there is no guarantee of the estimated CLEVER-Q score. Also, the advantage of CLEVER-Q over AC-rate is not discussed.


Minor issues:
- Appendix C.1, this is just huber loss instead of quantile huber loss.
- Appendix D, 'Refer to We'.


Questions:
1. If I understand correctly, the difference between DQN-CF and CIQ is just that the inteference loss does not propogate to the Q-network parameters in DQN-CF?
2. Depending on whether there are interference, how different are the outputs of f_2 and f_3? If f_2 output is similar to f_3 even when there is no interference, it would suggest that we only need f_2 and the switching mechanism can be removed.

Post rebuttal:
The authors have addressed most of my main concerns and provided detailed experiment results, hence I increase my score to 7.

---

> ### Author Response · Authors · 2020-11-20
> **Response to AnonReviewer3 (1/2)**
>
> We thank that the reviewer thinks our approach is novel and acknowledge our efforts on extensive experiment results. We also appreciate the reviewer for pointing out some presentation issues for us to improve the paper.
>
> ****
>
> **R3Q1** - “I believe that the proposed framework has the potential to enhance a single agent with multiple types of attacks. Is it possible to have multiple interference types during training?”
>
> **Ans** - We agree the proposed method can be extended to the setting of multiple types of attacks. Following the reviewer’s suggestion, we conduct an extension of CIQ dealing with multiple interferences (MI), CIQ-MI. The results are shown in Table G5 and Appendix E.6.
>
> Table G5. CIQ-MI: CIQ agent with an extended multi-interference (MI) architecture.
>
> | Train \ Test           | Gaussian | Adversarial | Gaussian + Adversarial |
> |------------------------|----------|-------------|------------------------|
> | Gaussian               | 195.1    | 154.2       | 96.3                   |
> | Adversarial            | 153.9    | 195.0       | 105.1                  |
> | Gaussian + Adversarial | 195.0    | 195.0       | 195.0                  |
>
> Below we show how the proposed CIQ model could be extended from the architecture shown in Figure 2. to the multi-interference (MI) setting. The design intuition is based on two-step inference by a **common encoder**, to infer a clean or noisy observation, followed by an **individual decoder** tied to an interference type, to infer noisy types and activate the corresponding Q-network (named $\theta_4$). Note that the two-step inference mechanism follows the Rubin's Causal Model (RCM) (Imbens & Rubin, 2010) as two sequential potential outcome estimation models (Rubin, 1974; Imbens & Rubin, 2010), where interfered observation $x^\prime_t$ is determined by two labels $i_{1,t}$ andi $i_{2,t}$  and $x^\prime_{t}= i_{1,t} (i_{2, t} \mathcal I_{1}(x_{t}) + (1 - i_{2, t})\mathcal I_{2}(x_{t}) )  + (1 - i_{1, t}) x_{t}$ extended from Eq. 1.
>
> As proof of concept, we consider two interference types together, Gaussian noise and adversarial perturbation. From the results shown in Table. G5, we find that the extended version of CIQ, CIQ-MI, is capable of making correct action to solve (over 195.0) the environment when training with mixed interference types (last row). Another finding is that robustness transferability (153.9/154.2) in CIQ-MI is slightly degraded compared to the transfer learning results (162.8/165.2) in Table G3. with the same training episodes ($500$) and runs ($20$), which could be caused by the increased requirement of model capacity (Ammanabrolu et al. 2019) of CIQ-MI.
>
> ****
>
> **R3Q2** - “The resulting agent would then have the potential to be resilient to all types of interferences during evaluation. Furthermore, it would then be possible to combine different types of adversarial attacks into adversarial training.”
>
> **Ans** - We agree that the adversarial training would be an interesting direction to enhance CIQ performance. Meanwhile, from the recent studies, applying adversarial training to deep models also needs careful designs as it may undermine model generalization (A. Raghunathan 2019; Su 2018). From our robustness transferability results in Table G2&G3, CIQ with adversarial training shows a competitive performance transferring to Gaussian noise but with a degraded performance transferring to the Blackout and Frozen conditions. We also add the results of CIQ trained with multiple interference types in Table G5 and Appendix E.6.
>
> ****
>
> **R3Q3** - “Causal Insight DQN other than an additional interference type output. As such, I am not very convinced by the causal inference insight”
>
> **Ans** - Predicting the interference type is indeed a key feature of CIQ, but the CIQ architecture is different from the regular DQN by incorporating causal inference insight from Rubin's Causal Model (RCM) (Rubin, 1974; Imbens & Rubin, 2010). We also motivate the design mechanism from both recent works from causal representation learning models and theoretical foundations in the updated section 3.3. and 3.4 (see the general response (1) and (2)). Intuitively, the switching mechanism (counterfactual inference) from RCM could be considered as a method to disentangle a single deep network into two non-parameter-sharing networks to improve model generalization under uncertainty. It has shown many advantages for representation learning in regression tasks (Shalit et al., 2017; Louizos et al., 2017). We show in our ablation study (Appendix E.3) that this mechanism using the predicted label is important for performance. We appreciate the reviewer’s suggestion for our presentation.
>
> ****

---

> > ### Author Response · Authors · 2020-11-20
> > **Response to AnonReviewer3 (2/2)**
> >
> > **R3Q4** - “Computation cost of CLEVER-Q is expensive, and there is no guarantee of the estimated CLEVER-Q score. Also, the advantage of CLEVER-Q over AC-rate is not discussed.”
> >
> > **Ans** - Thank you for pointing this out. We now realized that we did not fully explain the different roles CLEVER-Q and AC-rate play in robustness assessment, which could cause some confusion. In fact, their roles are actually complementary rather than conflicting. CLEVER-Q measures “sensitivity” in terms of the margin (minimum perturbation) required for a given state to change the original action. AC-rate measures “utility” in terms of the consistency (matching rate) between actions taken under interference v.s. without interference. However, unlike CLEVER-Q, AC-rate does not provide an assessment of how prone a state is to change in the presence of interference. On the other hand, unlike AC-rate, CLEVER-Q does not provide an assessment of how consistent the current policy (under interference) is when compared to the gold standard policy (no interference). Therefore, we believe both metrics are necessary in order to provide a comprehensive robustness assessment. We have updated our paper to better clarify their roles as complementary robustness metrics. In addition, we understand the reviewers’ concerns about the computation cost and the estimation quality of the CLEVER-Q score. However, we are not aware of any other interference-agnostic metric to measure the associated robustness in our setting, and that is what motivates us to delve into designing such a robustness metric. If there are any other metrics that the reviewer thinks will be useful to include, please let us know and we are happy to include it in our revised version.
> >
> > ****
> >
> > **R3Q5** - If I understand correctly, the difference between DQN-CF and CIQ is just that the interference loss does not propagate to the Q-network parameters in DQN-CF?
> >
> > **Ans** - Not propagating the interference loss to Q-network is indeed one key difference between CIQ and DQN-CF, but CIQ also has other architectural innovation as described in the answer to one of your previous questions (R3Q3).
> >
> > ****
> >
> > **R3Q6** - Depending on whether there is interference, how different are the outputs of f_2 and f_3? If f_2 output is similar to f_3 even when there is no interference, it would suggest that we only need f_2 and the switching mechanism can be removed.
> >
> > **Ans** - The difference in the outputs of f_2 and f_3 are actually large enough to affect the performance of the agent. We conducted ablation studies which show that the model variant without using f_3 (B2 in Appendix E.3) performs significantly worse than the complete CIQ model.
> >
> > Thank you again.

---

> > > ### Comment · AnonReviewer3 · 2020-11-24
> > > **Thanks for the authors' response**
> > >
> > > I thank the authors' for addressing most of my main concerns. Just as a quick verification, "Gaussian + Adversarial" in the newly added table 19. refers to either of the two attack types, instead of applying both of them (i.e. I_2(I_1(x))), is that correct? Also, what are the experiment settings of the architecture ablation studies? I am trying to compare it to one of the columns in other experiments to find out raw DQN and DQN-CF performances using the same settings.

---

> > > > ### Author Response · Authors · 2020-11-24
> > > > **The reviewer's comments are correct**
> > > >
> > > > **AQ1**: "Gaussian + Adversarial" in the newly added Table 19 refers to either of the two attack types, instead of applying both of them (i.e. $I_2$($I_1$(x))), is that correct?
> > > >
> > > > **Ans**:　Yes. Gaussian + Adversarial means every observation (state) can possibly undergo an interference with either Gaussian or Adversarial, but not their composite form. We also updated our paper to better clarify this setting.
> > > >
> > > > ****
> > > >
> > > > **AQ2**: What are the experiment settings of the architecture ablation studies?
> > > >
> > > > **Ans**: Appreciate you again for letting us know this missing description. The setting used for Table 16 is the same as the setting for the third column (noise level = 20%) in Table 5 and the third column (noise level = 20%) in Table 15, tested in Env$_1$ (Cartpole). We have updated the paper accordingly.
> > > >
> > > > ****
> > > >
> > > > Ｗe thank the reviewer for careful reading, and we are happy to answer any other question you may have.

---

### Official Review · AnonReviewer4 · 2020-10-28
**Interesting framework with promising experimental results**

**Rating:** 4
**Confidence:** 3

**Review:**

##########################################################################

Summary:

The paper presents a framework for deep reinforcement learning that is motivated by causal inference and with the central objective of being resilient to observational interferences. The key idea is to use interference labels in the training phase to learn a causal model including a hidden confounding state, and then use this model in the testing to make safer decisions and improve resilience. The authors also propose a new robustness measure, CLEVER-Q, which estimates a noise bound of an RL model below which the model's greedy decision would not be altered. The framework is tested extensively over multiple applications and under different types of observational interferences. The results show a clear advantage of the proposed framework over baseline RL methods in terms of resilience to interference.

##########################################################################

Reasons for Score:

The paper addresses an important problem in AI relating to the robustness of the algorithms and paradigms to noisy interference. The proposed framework appears to be sound and the experimental results show superior performance in comparison to other baseline RL methods. That being said, I am not very familiar with the literature on RL and Deep learning, so my decision is more of an educated guess. However, I do have a concern about the causal inference component of the paper (explained below) and this is reflected by the score.

##########################################################################

Cons:

The causal component of the proposed framework is not well-explained. More specifically, the causal graphical model in Figure 2 is introduced at the beginning of Subsection 3.3 very briefly, and the authors don't explain the intuition behind constructing this graph.
The authors say, "We use z_t to denote the latent state which can be viewed as a confounder in causal inference", but there is no explanation for why this makes sense. Is it just an assumption that happens to work?

Moreover, the following phrase appears to be inaccurate "knowing the interference labels it or not corresponds to different levels in Pearl’s causal hierarchy...: the intervention level with the interference labels and the association level without the information". Knowing vs not knowing the interference labels does not correspond to interventional vs associational levels in the causal hierarchy, but simply switching a variable/node in the CGM between observed and latent/unobserved. Such knowledge of a variable in the CGM does not account for an intervention.

##########################################################################

Questions during the rebuttal period:

It would help if the authors can clarify the issue raised in the "Cons" above regarding the clarity of the causal component and its central role, as claimed, in the proposed framework.

##########################################################################

Typos:

- p.1, "the RL agent is asked to learn a binary causation label and *embedded* a latent state into its model": embedded -> embed.
- p.3, "design an end-to-end structure ... and *evaluated* by treatment effects on rewards": The statement does not parse.
-p.3, "where M is a *fix* number for the history": fix -> fixed.
-p.3, "We assume that interference labels i_t *follows* an i.i.d. Bernoulli process": follows -> follow.

##########################################################################

Comments after Discussion:

I appreciate the effort made by the authors to elaborate on the causal formulation behind CIQ. However, the additional discussions in the paper are still confusing and raise soundness concerns. Some of the issues are discussed below.

1- Rubin's Causal Model: The authors reference RCM in Subsection 3.1 for the causal formulation yet the rest of the work does not seem to use the potential outcome notation. Instead, Subsection 3.3 uses graphical models and the do-operator which follows the causal framework by Pearl. Then, in Subsection 3.4, the authors go back to reference RCM. It is not clear why this alternation between the two approaches is employed.

2- If $z_t$ is defined as a function of $x_t$ and $i_t$, shouldn't the CGM reflect that with an arrow from $i_t$ to $z_t$ instead of it being the other way around in Fig.2(a)? Despite the attempt by the authors to elaborate on the causal formulation, I'm unable to map the structural equations such as Eq. (1) and the function of $z_t$ to the given CGM in Fig.2(a).

3- The discussion in Subsection 3.3 leading to Eq. (3) sounds flawed to me. Quoting the authors, "the interference model of Eq. (1) can be viewed as the intervention logic with the interference label it being the treatment information". This statement is elaborating on the formulation of $x_t'$ where $x_t$ is intervened on and replaced by an interfered state when $i_t=1$. Alternatively, $x_t$ is kept intact when $i_t=0$. This intervention on the mechanism of $x_t$ happens whether we obtain $i_t$ and train the DQN with it or not. In this sense, the intervention is not happening under the CIQ framework only, but also when we simply train based on $x_t'$. Accordingly, it is not clear to me how "the learning problem is elevated to Level II of the causal hierarchy" due to the presence of the interference labels. To be clear, I'm not questioning the significance of using the interference labels in the training, but rather the causal story and formulation behind CIQ.

---

> ### Author Response · Authors · 2020-11-20
> **Response to AnonReviewer4**
>
> We thank the reviewer for acknowledging the importance and soundness of our work. We also appreciate the reviewer for pointing out some presentation issues for us to improve the paper.
>
> ****
>
> **R4Q1** - “the causal graphical model in Figure 2 is introduced at the beginning of Subsection 3.3 very briefly“
>
> **Ans** - Following your comment, we add more sections to clarify the connection between causal inference, design motivation, explanation of theoretical foundation from references, as summarized in the general response (1) and (2) in section 3.3 and 3.4.
>
> ****
>
> **R4Q2** - “explanation for confounder z_t”
>
> **Ans** - Using the causal graphical model defined in Figure 2(a), the confounder can be formally defined by $z_t = h(x_t, i_t)$ where $h$ is a compression function such that $z_t$ is a hidden confounder in the CGM. Note that such function $h$ exists by simply choosing $h$ to be the identity function. What we assume is that there is a compression function $h$ such that $z_t$ is low-dimensional, and similar to (Louizos et al., 2017), we aim to learn to predict this low-dimensional hidden confounder by a neural network. We add this discussion in a new paragraph in Section 3.3 (see general response).
>
> ****
>
> **R4Q3** - “Knowing vs not knowing the interference labels does not correspond to interventional vs associational levels in the causal hierarchy, but simply switching a variable/node in the CGM between observed and latent/unobserved. Such knowledge of a variable in the CGM does not account for an intervention.”
>
> **Ans** - To further clarify our descriptions and make connections to causal hierarchy, we add a new paragraph in Section 3.3 (see general response) with more details on the causal hierarchy (see Table G1) and what information is available for each level. We hope the explanation addresses the reviewer’s concern.
>
> Table. G1 Causal Hierarchy in our resilient DRL settings.
>
> | Level                         | Activity    | Symbol                   | Example    |
> |-------------------------------|-------------|--------------------------|------------|
> | ($\mathbb{I}$)  Association   | Observing   | $P(r_t \| x'_t) $         | DQN        |
> | ($\mathbb{II}$)  Intervention | Intervening | $P(r_t\| do(x'_t), i_t)$ | CIQ (ours) |
>
> ****
>
> **R4Q4** -Typos Issue.
>
> **Ans** - We have fixed the typos issues mentioned by the reviewer. Many thanks!

---

### Official Review · AnonReviewer2 · 2020-10-30
**Interesting paper and relevant approach, supported by rigorous experimentation. On the downside, the novelty of the paper does not seem major and the predefined nature of interventions might make it unrealistic in a lot of RW scenarios.**

**Rating:** 7
**Confidence:** 3

**Review:**

Overview: The paper introduces a causal mechanism that both creates and explains away noise interventions into observational data fed into RL agents. The authors propose a form of resilient agent, that based on training data containing labeled interventions, learns both Q function and the causal impact of interventions on the Q function. The model architecture consists of a intervention predictor and a split parametrization of the Q function estimation as a function of intervention as shown in L^{CIQ} presented as eq.3. Finally, the authors show the performance of their proposed method on 4 visual based RL agents with two types of interventions (attacks): namely adversarial and blackout against classical baselines such as DQN and DQN with safe actions.

Pros:
- Clarity: Overall I find the paper well-written and reasonably easy to follow. The problem is well motivated and the related work relevant.
- Significance/Impact: I think the problem that the paper is trying to solve is relevant and with potentially big impact
- Experimental design: I think the experiments section is relevant and makes a strong case for the method

Cons:
- Though clear at most times, the paper should spend more time explaining the basic causal terminology and the assumptions behind the causal graph introduced in Figure 2a. I find that the authors introduce the causal coneepts in an informal, intuitive way, but that should be followed-up by a clear formalism.
- I find that the biggest downside of the method is that it needs to be trained with the type of invervention that the agent will be resilient to. It would be interesting to create an intervention detector that is fully unsupervised and that is based more on a state anomaly detection.

Bordeline:
- Novelty: In terms of novelty, the paper is a relatively straight-forward application of do-calculus to Q-value learning.

Final comments:
	Overall i found the paper interesting and the approach relevant, supported by rigurous experimentation. On the downside, the novelty of the paper does not seem major and the predefined nature of interventions might make it unrealistic in a lot of RW scenarios.

---

> ### Author Response · Authors · 2020-11-20
> **Response to AnonReviewer2**
>
> We thank the reviewer for acknowledging our work to be novel and the efforts on the experiments, and we also appreciate the useful suggestions on the presentation.
>
> ****
>
> **R2Q1** - “causal terminology and the assumptions behind the causal graph in Figure 2a.”
>
> **Ans** - We add a new paragraph in Section 3.3 (see general response) in the revised paper to clarify the causal terminology and the assumptions behind the causal graph.
>
> ****
>
> **R2Q2** - “… needs to be trained with the type of intervention that the agent will be resilient to. It would be interesting to create an intervention detector that is fully unsupervised and that is based more on a state anomaly detection …”
>
> **Ans** - We agree with the comment of extending CIQ to much complex and unsupervised conditions. Our original experiment design is based on careful condition control and intervened observations in order to clearly demonstrate the gain in our model design. Based on R2’s comments, we conduct two additional experiments to study (1) robustness transferability among different interference types (see Table G2&G3); and (2) the performance of CIQ against two different interference types during training and testing (see Table G5). More Details are given in the general summary and Appendix E.5.
>
> Table. G2. DQN adaptation: train and test on different interference (noise) in Env$_1$.
>
> | Train\Test | Gaussian | Adversarial| Blackout| Frozen |
> |:-----------------------|:-------------------:|:----------------------:|:-------------------:|-----------------:|
> | Gaussian              | 67.4              | 38.4                 | 43.7              | 52.1            |
> | Adversarial           | 53.2              | 42.5                 | 35.3              | 44.2            |
> | Blackout              | 46.2              | 27.4                 | 85.7              | 50.3            |
> | Frozen                | 62.3              | 26.2                 | 45.9              | 62.1            |
>
> Table G3. CIQ adaptation: train and test on different interference (noise) in Env$_1$.
>
> | Train\Test | Gaussian| Adversarial| Blackout | Frozen|
> |-----------------------|-------------------|----------------------|-------------------|-----------------|
> | Gaussian              | 195.1    | 165.2                | 158.2             | 167.8           |
> | Adversarial           | 162.8             | 195.0       | 152.4             | 162.5           |
> | Blackout              | 131.3             | 121.1                |195.3    | 145.7           |
> | Frozen                | 161.6             | 135.8                | 147.1             | 195.2  |
>
> Table G5. CIQ-MI: CIQ agent with an extended multi-interference (MI) architecture.
>
> | Train \ Test           | Gaussian | Adversarial | Gaussian + Adversarial |
> |------------------------|----------|-------------|------------------------|
> | Gaussian               | 195.1    | 154.2       | 96.3                   |
> | Adversarial            | 153.9    | 195.0       | 105.1                  |
> | Gaussian + Adversarial | 195.0    | 195.0       | 195.0                  |
>
> ****
>
> **R2Q3** - “On the downside, the novelty of the paper does not seem major and the predefined nature of interventions might make it unrealistic in a lot of RW scenarios.”
>
> **Ans** - We agree that knowing all types of interventions that could happen in RW can be challenging. However, this challenge should not prevent us from exploiting known intervention types to train resilience machine learning models, in order to reduce the gap between simulation and practical deployment. To further illustrate our point, we currently consider the same interference type in training and testing, but they can be different in distribution and dynamics (see Ans. to R2Q2 for discussion).
>
> ****
>
> **R2Q4** - Minor Clarification: “two types of interventions (attacks) … ”
>
> **Ans** - We would like to point out that “four” types of interventions have been evaluated in this work as discussed in section 3.1. More experiment results are shown in Appendix C.
>
> Thank you for the suggestion again.

---

### Author Response · Authors · 2020-11-20
**General Response and Summary of Main Updates (1/2)**

We would like to thank all reviewers for spending time and effort to read our submission and provide valuable suggestions and comments on this work.

With this revised version and the additional page in the main content, we make updates from three main perspectives and highlight them in red color:

****

**(1)** In Sections 3.3, we provide the following connection of causal learning terminology to our model design intuition and theoretical foundations.

Table. G1 Causal Hierarchy in our resilient DRL settings.


| Level                         |    Activity    | Symbol                   | Example    |
|:-------------------------------|:-------------:|:--------------------------:|------------:|
| ($\mathbb{I}$) Association   |     (Observing)   | $P(r_t \| x'_t) $         | DQN        |
| ($\mathbb{II}$) Intervention |     (Intervening) | $P(r_t\| do(x'_t), i_t)$ | CIQ (ours) |

Formally, we define $z_t = h(x_t, i_t)$ to be the hidden confounder. Here $h$ is a function that compresses $(x_t, i_t)$ into a confounder such that the causal graphical model (CGM) holds. It is clear from Eq. (1) and the MDP definition that the CGM holds with $h$ being the identity function, i.e., $z_t = (x_t, i_t)$. We assume that there exists some unknown compression function $h$ such that $z_t$ is low-dimensional. Similar to TARNets (Shalit et al., 2017; Louizos et al., 2017), we aim to learn to predict this low-dimensional hidden confounder by a neural network.

According to the CGM, different training settings correspond to different levels of Pearl's **causal hierarchy** (Pearl 2009; Pearl 2019; E Bareinboim et al. 2020) as shown in Table 1. If only the observations are available, the training process corresponds to Level $\mathbb{I}$ of the causal hierarchy, which associates the outcome $r_t$ to the input observation $x'_t$ directly by $P(r_t|x'_t)$.On the other hand, when interference type $\mathcal I$ and the interference labels $i_t$ are available during training, the learning problem is elevated to Level $\mathbb{II}$ of the causal hierarchy. In particular, the interference model of Eq. (1) can be viewed as the intervention logic with the interference label $i_t$ being the treatment information. With this information, we can describe the causal inference problem by
$P(r_t| do(x'_t), i_t)=P(r_t| F^{\mathcal I}(x_t, i_t)=x'_t, i_t)$
with the do-operator (Pearl 2019) in the intervention level of the causal hierarchy.


****

**(2)** In Section 3.4, we provide the following design intuition from Rubin’s Causal Model

The design intuition of our inference mechanism is based on the potential outcome estimation theory in Rubin’s Causal Model (RCM) (Rubin, 1974; Imbens & Rubin, 2010) and modeling of the interference scenario as described in Eq. 1. Intuitively, the switching mechanism (counterfactual inference (Imbens & Rubin, 2010; Shalit et al., 2017; Louizos et al., 2017) from RCM could be considered as a method to disentangle a single deep network into two non-parameter-sharing networks to improve model generalization under uncertainty. It has shown many advantages for representation learning in regression tasks (Shalit et al., 2017; Louizos et al., 2017). We also provide more implementation details in appendix C.1.


****

---

> ### Author Response · Authors · 2020-11-20
> **General Response and Summary of Main Updates (2/2)**
>
> **(3)**  Following reviewers’ suggestions, we add three additional experiments:
>
> **I.** Robustness transferability among different interference types (Appendix E.5)
>
> Table. G2. DQN adaptation: train and test on different interference (noise) in Env$_1$.
>
> | Train\Test | Gaussian | Adversarial| Blackout| Frozen |
> |:-----------------------|:-------------------:|:----------------------:|:-------------------:|-----------------:|
> | Gaussian              | 67.4              | 38.4                 | 43.7              | 52.1            |
> | Adversarial           | 53.2              | 42.5                 | 35.3              | 44.2            |
> | Blackout              | 46.2              | 27.4                 | 85.7              | 50.3            |
> | Frozen                | 62.3              | 26.2                 | 45.9              | 62.1            |
>
> Table G3. CIQ adaptation: train and test on different interference (noise) in Env$_1$.
>
> | Train\Test | Gaussian| Adversarial| Blackout | Frozen|
> |-----------------------|-------------------|----------------------|-------------------|-----------------|
> | Gaussian              | 195.1    | 165.2                | 158.2             | 167.8           |
> | Adversarial           | 162.8             | 195.0       | 152.4             | 162.5           |
> | Blackout              | 131.3             | 121.1                |195.3    | 145.7           |
> | Frozen                | 161.6             | 135.8                | 147.1             | 195.2  |
>
> ****
>
> **II** A new baseline in architecture ablation study of CIQ  (Appendix E.3)
>
> Table G4. Structure-wise ablation studies.
>
> | Model                                             | Return | CLEVER-Q | AC-Rate |
> |---------------------------------------------------|--------|----------|---------|
> | CIQ                                               | 195.1  | 0.241    | 97.3    |
> | B3: CIQ w/o providing grounded $i_t$ for training | 135.1  | 0.142    | 53.6    |
>
> ****
>
> **III** Extension of CIQ model to a multiple-interference setting (Appendix E.6)
>
> Table G5. CIQ-MI: CIQ agent with an extended multi-interference (MI) architecture.
>
> | Train \ Test           | Gaussian | Adversarial | Gaussian + Adversarial |
> |------------------------|----------|-------------|------------------------|
> | Gaussian               | 195.1    | 154.2       | 96.3                   |
> | Adversarial            | 153.9    | 195.0       | 105.1                  |
> | Gaussian + Adversarial | 195.0    | 195.0       | 195.0                  |
>
> We then make separate responses to address each reviewer’s comments.

---

### Author Response · Authors · 2020-11-23
**Looking forward to reviewers' reply**

Dear reviewers,

As Discussion Stage 2 is about to end, we are looking forward to your reply.

Please kindly let us know if our response has addressed your initial questions. We appreciate your input and are happy to discuss any follow-up questions. Thank you again!

---

### Decision · Program_Chairs · 2021-01-07
**Final Decision**

**Decision:**

Reject

**Comment:**

This paper presents a deep reinforcement learning method that aims at ensuring resilience to observational interference. During training labels that indicate presence or absence of interference are available to the algorithm. The training objective is augmented to learn the prediction of interference that is used at test time to infer the interference label. The experimental results show superior performance in comparison to other baseline RL methods.

The main objection raised by the reviewers was on the confusing and possibly unsound causal formulation. The authors' clarifications during the discussion did not eliminate bur rather exacerbated the reviewers' doubts. I read the paper in full myself to understand whether the reviewers' confusion was justified, and whether it could be easily resolved by an improved explanation, or it is a more serious issue.  I did not succeed in clearly understanding the causal formulation nor its relevance, and also have  soundness concerns. Figure 2a does not seem to be a correct explanation of the causal mechanism. It is also not clear from this figure why z is called confounder. More generally, I was not able to reach a coherent and sound causal formulation from the authors' explanation. My conclusion is that the framing of the paper as causal inference based is not well justified.